# How oscillating aerodynamic forces explain the timbre of the hummingbird's hum and other animals in flapping flight

**Ben J Hightower[1†], Patrick WA Wijnings[2†], Rick Scholte[3], Rivers Ingersoll[1], Diana D Chin[1], Jade Nguyen[1], Daniel Shorr[1], David Lentink[1]\***

[1]Mechanical Engineering, Stanford University, Stanford, United States; [2]Electrical Engineering, Eindhoven University of Technology, Eindhoven, Netherlands; [3]Engineering, Sorama, Eindhoven, Netherlands

**Abstract** How hummingbirds hum is not fully understood, but its biophysical origin is encoded in the acoustic nearfield. Hence, we studied six freely hovering Anna's hummingbirds, performing acoustic nearfield holography using a 2176 microphone array in vivo, while also directly measuring the 3D aerodynamic forces using a new aerodynamic force platform. We corroborate the acoustic measurements by developing an idealized acoustic model that integrates the aerodynamic forces with wing kinematics, which shows how the timbre of the hummingbird's hum arises from the oscillating lift and drag forces on each wing. Comparing birds and insects, we find that the characteristic humming timbre and radiated power of their flapping wings originates from the higher harmonics in the aerodynamic forces that support their bodyweight. Our model analysis across insects and birds shows that allometric deviation makes larger birds quieter and elongated flies louder, while also clarifying complex bioacoustic behavior.

**\*For correspondence:**
dlentink@stanford.edu

[†]These authors contributed equally to this work

## Introduction

Birds, bats, and insects flap their wings to generate unsteady aerodynamic forces that lift their body into the air, which enables them to fly. When their flapping wings move through air, they create unsteady pressure fluctuations that radiate outward at the speed of sound. In addition to furnishing flight, pressure waves serve various acoustic communication functions during behavioral displays. Male *Drosophila* use aerodynamically functional wings to create humming songs near their flapping frequency to increase female receptivity to mating (*von Schilcher, 1976*). In a more sophisticated form of courtship behavior, male and female mosquitoes duet at the third harmonic (multiple) of their wingbeat frequency (*Cator et al., 2009*). In contrast, pigeons use modified primary feathers that sonate around 1 kHz when they start flapping their wings that incite flock members to fleeing and take-off behavior (*Davis, 1975*; *Hingee and Magrath, 2009*; *Niese and Tobalske, 2016*; *Murray et al., 2017*). Feather sonation during flapping flight may also communicate information like flight speed, location in 3D space, and wingbeat frequency to conspecifics (*Larsson, 2012*). Hence, male broad-tailed hummingbirds generate a whistling sound with modified primary feathers in their flapping wings during displays to defend courting territories (*Miller and Inouye, 1983*). Silent fliers like owls, on the other hand, suppress the aerodynamic sound generated by their wings to mitigate interference with their hearing and escape prey detection (*Geyer et al., 2013*; *Jaworski and Peake, 2020*; *Kroeger et al., 1972*; *Sarradj et al., 2011*; *Clark et al., 2020*). Their flapping wings also generate less structural noise (*Clark et al., 2020*) because their feathers lack the noisy directional fastening mechanism that locks adjacent flight feathers during wing extension in other bird species (*Matloff et al., 2020*). These diverse adaptations illustrate how a wide range of mechanisms can contribute to the sound that flapping wings generate. Consequently, it is not fully understood how

**eLife digest** Anyone walking outdoors has heard the whooshing sound of birdwings flapping overhead, the buzzing sound of bees flying by, or the whining of mosquitos seeking blood. All animals with flapping wings make these sounds, but the hummingbird makes perhaps the most delightful sound of all: their namesake hum. Yet, how hummingbirds hum is poorly understood.

Bird wings generate large vortices of air to boost their lift and hover in the air that can generate tones. Further, the airflow over bird wings can be highly turbulent, meaning it can generate loud sounds, like the jets of air coming out of the engines of aircraft. Given all the sound-generating mechanisms at hand, it is difficult to determine why some wings buzz whereas others whoosh or hum.

Hightower, Wijnings et al. wanted to understand the physical mechanism that causes animal wings to whine, buzz, hum or whoosh in flight. They hypothesized that the aerodynamic forces generated by animal wings are the main source of their characteristic wing sounds. Hummingbird wings have the most features in common with different animals' wings, while also featuring acoustically complex feathers. This makes them ideal models for deciphering how birds, bats and even insects make wing sounds.

To learn more about wing sounds, Hightower, Wijnings et al. studied how a species of hummingbird called Anna's hummingbird hums while drinking nectar from a flower. A three-dimensional 'acoustic hologram' was generated using 2,176 microphones to measure the humming sound from all directions. In a follow-up experiment, the aerodynamic forces the hummingbird wings generate to hover were also measured. Their wingbeat was filmed simultaneously in slow-motion in both experiments. Hightower, Wijnings et al. then used a mathematical model that governs the wing's aeroacoustics to confirm that the aerodynamic forces generated by the hummingbirds' wings cause the humming sound heard when they hover in front of a flower. The model shows that the oscillating aerodynamic forces generate harmonics, which give the wings' hum the acoustic quality of a musical instrument.

Using this model Hightower, Wijnings et al. found that the differences in the aerodynamic forces generated by bird and insect wings cause the characteristic timbres of their whines, buzzes, hums, or whooshes. They also determined how these sounds scale with body mass and flapping frequency across 170 insect species and 80 bird species. This showed that mosquitos are unusually loud for their body size due to the unusual unsteadiness of the aerodynamic forces they generate in flight.

These results explain why flying animals' wings sound the way they do – for example, why larger birds are quieter and mosquitos louder. Better understanding of how the complex forces generated by animal wings create sound can advance the study of how animals change their wingbeat to communicate. Further, the model that explains how complex aerodynamic forces cause sound can help make the sounds of aerial robots, drones, and fans not only more silent, but perhaps more pleasing, like the hum of a hummingbird.

flapping wings generate their characteristic sound—from the mosquito's buzz, the hummingbird's hum, to the larger bird's whoosh.

Our physical understanding of how wings generate sound is primarily based on aircraft wing and rotor aeroacoustics (*Brooks and Pope, 1989*; *Crighton, 1991*). In contrast to animals, however, engineered wings do not flap, do not change shape dynamically, are much larger, and operate at much higher speeds (higher Reynolds numbers). They also operate at lower angles of attack to avoid stall, which results in more compact airflow patterns than animals generate in flapping flight (*Ellington et al., 1996*; *Dickinson et al., 1999*; *Mueller, 2001*). Despite these marked differences, rotors and flapping wings have one thing in common: they both revolve around a center pivot. Whereas flapping wings reciprocate along the joint, rotors revolve unidirectionally. The revolution of rotors generates loud tonal noise, because the pressure field they generate rotates in space at the same frequency (*Lighthill, 1954*; *Lowson, 1965*). Similarly, when animals flap their wing back and forth along the shoulder joint during each stroke, they create a high-pressure region below their wing and a low-pressure region above. The pressure differences are associated with the wing's high lift and drag, respectively (*Sane and Dickinson, 2001*; *Wu and Sun, 2004*). Computational fluid

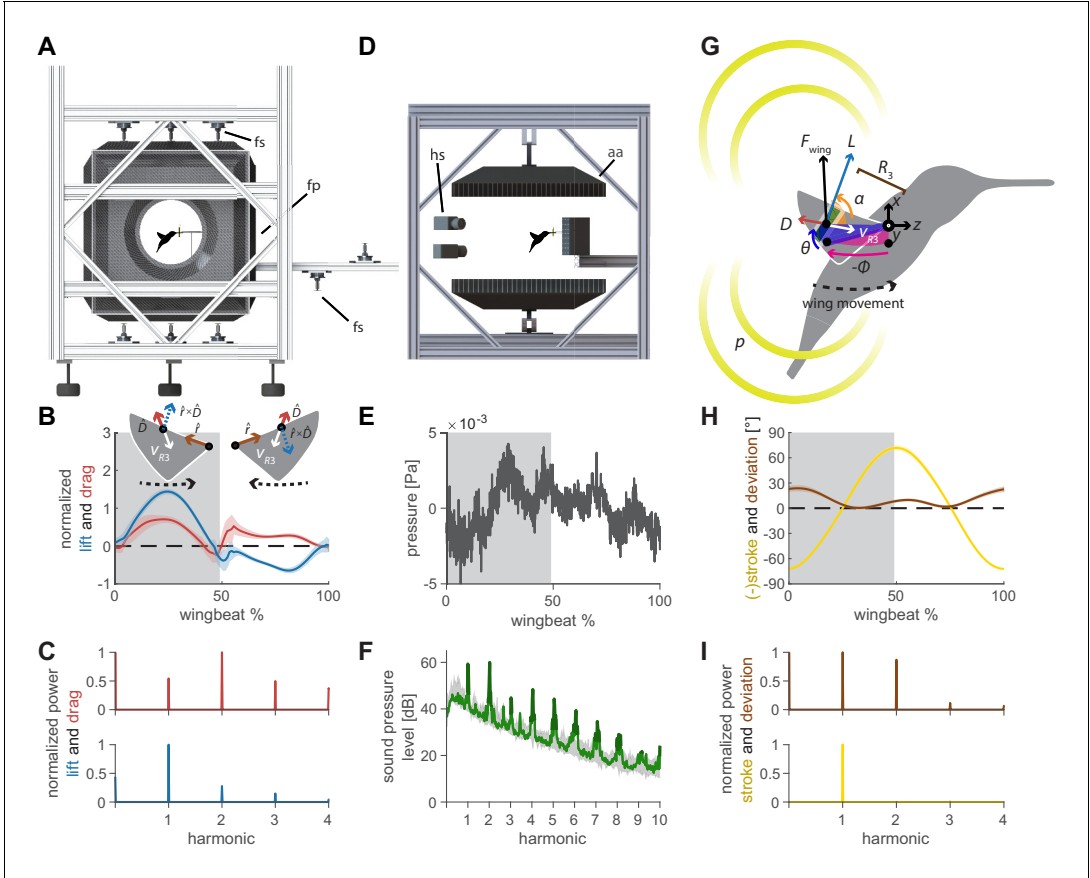

**Figure 1.** Oscillating aerodynamic force and acoustic field measurements to determine how hummingbirds hum. (**A**) 3D aerodynamic force platform setup to measure the forces generated by a hovering hummingbird. Each of the flight arena's walls comprises a force plate (fp) instrumented by three force sensors (fs), two additional force sensors instrument the perch. The six DLT calibrated cameras imaging through three orthogonal ports in pairs are not shown. (**B**) The lift and drag force generated by hovering hummingbirds during a wingbeat (gray area, downstroke; mean ± std based on $N = 6$ birds, each bird made two flights, $n = 5$ wingbeats were fully analyzed per flight for 60 total wingbeats). Lift is negative during the upstroke since the direction of the lift vector is perpendicular to the wing velocity while the drag vector is parallel and opposite to the wing velocity direction, resulting in the lift vector being defined as the cross product of the wing velocity direction and the drag direction (inset). (**C**) Most of the frequency content in the lift profile is contained in the first harmonic and corresponds to the high forces generated during downstroke (first harmonic mean ± standard deviation is $44.2 ± 1.8$ Hz across all birds and flights). In contrast, the frequency content in the drag profile is contained primarily in the second harmonic and corresponds to the equivalent drag generated during the up and downstroke. (**D**) Acoustic flight arena in which hovering hummingbirds ($N = 6$ birds, $n = 2$ flights per bird) were surrounded by four acoustic arrays (labeled aa; $2 ×1024$ and $2 × 64$ microphones) and four high-speed cameras (hs) while feeding from a stationary horizontal flower (separate experiment with six other individuals). (**E**) Throughout a wingbeat, each microphone records the local acoustic field generated by the hovering hummingbird (microphone located at the center above bird #1). (**F**) To generate a representative spectrum of a single bird, the signals of all microphones in all arrays around the bird were summed (green line: $N = 1$, $n = 1$) and plotted up to the tenth harmonic. The background spectrum of the lab (range over all trials) is plotted in gray, showing the hum consists primarily of tonal noise higher than the background at wingbeat harmonics (dark green line, 3 dB above maximum background noise). In addition, several smaller non-harmonic tonal peaks can be observed between the first and fourth harmonic with a dB level equivalent to the sixth - seventh harmonic. (**G**) To determine the acoustic source of the hum, we constructed a simple model that predicts the acoustic field. The acoustic waves radiate outwards from the overall oscillating force ($F_{wing}$) generated by each wing, which can be decomposed into the lift ($L$) and drag ($D$) forces generated by each wing (recorded in vivo, B). To predict the aeroacoustics, these forces are positioned at the third moment of inertia of the wing ($R_3$) and oscillate back and forth due to the periodic flapping wing stroke ($\varphi$) and deviation angle ($\theta$) (recorded in vivo, H). Angle of attack $\alpha$ is defined for modeling flapping wing hum across flying species (**Figure 4**). (**I**) Hummingbird wing kinematics ($\varphi$, $\theta$) measured in vivo from the 3D aerodynamic force platform experiment (gray area, downstroke; mean ± std based on $N = 6$ birds, $n = 2$ flights). (**I**) Whereas most of the frequency content in the stroke profile is contained in the first harmonic, the content in the deviation profile extends to the second and third harmonics.

dynamics (CFD) simulations of flapping insect wings suggest that the acoustic field can be characterized as a dipole at the wingbeat frequency (*Bae and Moon, 2008*; *Geng et al., 2017*; *Seo et al., 2019*). Further, flapping wing pitch reduction (*Nedunchezian et al., 2018*) and increased wing

flexibility (*Nedunchezian et al., 2019*) reduces the simulated nearfield sound pressure level. All these findings point to the potential role of oscillating aerodynamic forces in generating wing hum. Indeed, numerical simulation of the Ffowcs Williams and Hawkings aeroacoustic equation (*Williams and Hawkings, 1969*) showed that the farfield hum of flapping mosquito wings is primarily driven by aerodynamic force fluctuation (*Seo et al., 2019*). Despite these important advances, in vivo acoustic near-field measurements are lacking. Finally, there is no simple model that can satisfactorily integrate flapping wing kinematics and aerodynamic forces to predict the acoustic near and far field generated by animals across taxa without using computationally expensive fluid dynamic simulations.

Hummingbirds are an ideal subject for developing and testing a model of flapping wing hum: their wing kinematics and unsteady aerodynamic forces are very repeatable during hover (*Altshuler and Dudley, 2003*; *Tobalske et al., 2007*; *Ingersoll and Lentink, 2018*). Further, hummingbird wing morphology and flight style share similarities with both birds and insects. In addition to high-frequency feather sonations, hummingbirds produce a prominent hum that is qualitatively similar to an insect's buzz. Earlier aeroacoustics studies of hummingbirds have resolved the farfield acoustic pressure field at a distance greater than 10 or more body lengths away from the hummingbird (*Clark and Mistick, 2018a*; *Clark, 2008*; *Clark et al., 2016*; *Clark and Mistick, 2018b*). While this distance relates to how humans perceive and interact with these animals, hummingbirds frequently interact with conspecifics and other animals at more intimate distances—in the acoustic nearfield. Furthermore, wing hum can announce a hummingbird's presence, especially to the opposite sex (*Hunter, 2008*). Although their audiogram has yet to be established below 1 kHz (*Pytte et al., 2004*), this and other behavioral evidence suggests hummingbirds may be able to perceive the wing hum from a conspecific. Finally, the hum may reveal the hummingbird's presence to predators in plant clutter when vision is obstructed.

To resolve how the oscillating aerodynamic force generated by flapping wings may contribute to wing hum, we developed a new aerodynamic force platform (*Ingersoll and Lentink, 2018*; *Lentink et al., 2015*; *Hightower et al., 2017*) to directly measure the net 3D aerodynamic force generated by freely hovering hummingbirds. We integrated this data in a new aeroacoustics model to predict the sound radiated due to the oscillating forces from flapping wings. Our model is idealized in the sense that it assumes the wings are rigid airfoils, thereby neglecting auxiliary effects such as wingtip flutter, feather whistle and (turbulent) vortex dynamics. Next, we compared the predicted acoustic field with novel acoustic nearfield recordings for six freely hovering hummingbirds, which corroborates the predictive power of our minimal model. We then used our validated model to determine how flapping wing hum depends on the frequency content in the oscillating forces across mosquitos, flies, hawkmoths, hummingbirds, and parrotlets in slow hovering flight. Finally, we used these findings to determine how the hum scales with body mass and flapping frequency across 170 insect and bird species.

## Results

### In vivo 3D aerodynamic force and acoustic nearfield measurements

To determine how the flapping wings of hovering hummingbirds generate unsteady aerodynamic forces as well as their namesake acoustic humming signature, we combine aerodynamic force platform (*Figure 1A*) and microphone array recordings (*Figure 1D*) in vivo. The aerodynamic force platform integrates both the steady and unsteady components of the pressure field around the bird up to three times the wingbeat frequency, which are associated with its net 3D aerodynamic forces. In contrast, the microphone arrays measure the unsteady component of the pressure field around the bird up to ~1000 times the wingbeat frequency (of which we studied the first ten harmonics): the acoustic field. Critically, these two representations of the pressure fluctuations generated by the bird should relate mechanistically if the acoustic field of the hummingbird's hum originates primarily from the oscillating aerodynamic lift and drag forces generated by the flapping wings.

The oscillating 3D aerodynamic forces were recorded simultaneously with the wingbeat kinematics using three calibrated stereo high-speed camera pairs (*Figure 1A*; $N$ = 6 birds, $n$ = 2 flights per bird, $n$ = 5 wingbeats per flight: 60 total wingbeats). We combined the 3D aerodynamic forces, 3D wing kinematics and wing morphology measurements to decompose the oscillating lift and drag

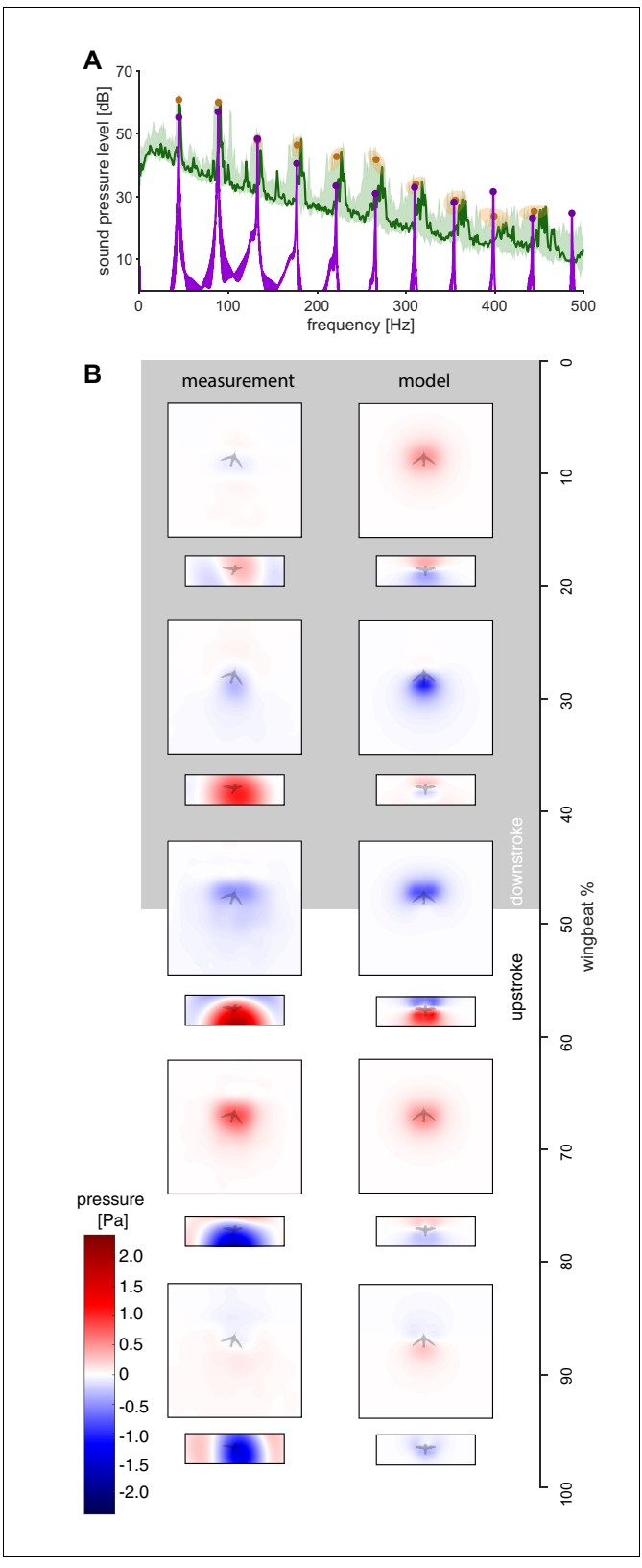

**Figure 2.** The measured spectra and holograms match those predicted by the simple aeroacoustics model. (**A**) A representative acoustic spectrum measured from all arrays for hummingbird #1 in hover is shown in dark green (*n* = 1), while the range for *N* = 6 hummingbirds is shown in light green. The variation in the frequency and sound pressure level (SPL) peak value associated with each harmonic is shown with orange circles (mean) and ellipsoids

*Figure 2 continued on next page*

*Figure 2 continued*

(width and height, 68% confidence intervals; their asymmetric shape stems from computing the covariance in Pascals while the spectrum is in dB). The peak sound pressure levels predicted by our acoustic model (purple line) match those of the measured spectrum up to higher harmonics. In addition, several smaller non-harmonic tonal peaks can be observed between the first and fourth harmonic with a dB level equivalent to the sixth - seventh harmonic. The predicted spectrum starts at the numerical noise floor, of which the amplitude ($< -10$ dB) is physically irrelevant. (B) Acoustic holograms throughout the example wingbeat for hummingbird #1 (*Figure 1E,F*) are presented side-by-side as measured (left) and modeled (right) for the top and front array microphone positions. There is reasonable spatial and temporal agreement between the measured and predicted acoustic nearfield centered around stroke transition (30–70%) where the pressure transitions from minimal (blue) to maximal (red).

The online version of this article includes the following figure supplement(s) for figure 2:

**Figure supplement 1.** Spectra and holograms show agreement between a 10-element distributed source model and the equivalent single point source model.

**Figure supplement 2.** Spectra shows good agreement between full model and simplified model.

**Figure supplement 3.** Illustration of regularization for lift and drag.

**Figure supplement 4.** Choice of regularization constants does not appreciably affect smoothing.

**Figure supplement 5.** Spectra of all inputs into the acoustic model reveal the source of humming harmonics and evidence of frequency mixing.

**Figure supplement 6.** Agreement between calculated lift and weight support in the vertical direction.

**Figure supplement 7.** Sensitivity of the spectrum to the location of the acoustic point force along the wing radius.

forces that each wing generates throughout the wingbeat (*Figure 1B,C*). The oscillating lift trace consists primarily of the peak force generated during downstroke, which corresponds to a peak in its spectrum at the first wingbeat harmonic (44.2 ± 1.8 Hz). The drag trace consists of two equivalent drag peaks during the upstroke and downstroke, which corresponds to a dominant peak in its spectrum at the second harmonic. We also measured the 3D beak contact force on the artificial flower from which the hummingbird was feeding, which is negligible (5.2 ± 2.3% bodyweight).

The 3D acoustic field associated with the bird's hum was quantitatively reconstructed from measurements recorded in a custom flight arena using four acoustic arrays (*Figure 1D*; $N = 6$ birds, $n = 18$ flights total, see *Supplementary file 1* for details). The recording by a single microphone centered above the bird shows a typical pressure trace throughout a single wingbeat (*Figure 1E*). The many fluctuations explain the rich frequency content revealed in the acoustic spectrum averaged over all microphones (*Figure 1F*). These include strong peaks at the fundamental frequencies of the wingbeat as well as its higher harmonics, which rise prominently above the background noise floor and characterize the hummingbird hum.

## Aeroacoustics model of the hum synthesizes in vivo forces and wing kinematics

To determine if the low frequency oscillating forces generated by the birds' flapping wings drive the characteristic humming sound spectrum, we develop a simple aeroacoustics model based on the governing acoustics equations that predict the resulting acoustic field. Our minimal model of the acoustic pressure field radiated by the flapping wings (*Figure 1G*) depends only on the physical properties of air, the wing stroke kinematics (*Figure 1H*), and the oscillating lift and drag forces that we measured in vivo (*Figure 1B*).

Aerodynamic analysis of propellers shows how a radial force distribution can be integrated and represented by the net force at the center of pressure, a characteristic radial location where the net force acts (*Weis-Fogh, 1973*). Analogously, we determine that the acoustic sound radiation of an unsteady aerodynamic force distribution over the wing can also be concentrated into an equivalent point force at the effective acoustic source location along the wing, similar to propeller noise theory (*Lowson, 1965*). The effective radius of this point, measured with respect to the shoulder joint, is equal to the point at which the net drag force results in the same net torque on the wing (*Lowson, 1965*). This radius lies at the wing-length-normalized third moment of area for flapping wings, $R_3/R$ (*Weis-Fogh, 1973*). For Anna's hummingbirds $R_3/R$ is equal to 55% wing radius (*Kruyt et al., 2014*). In practice, the effective radius for acoustic calculations can differ somewhat from the effective radius for a point force (*Lowson, 1965*). Therefore, we conduct a dimensional analysis to

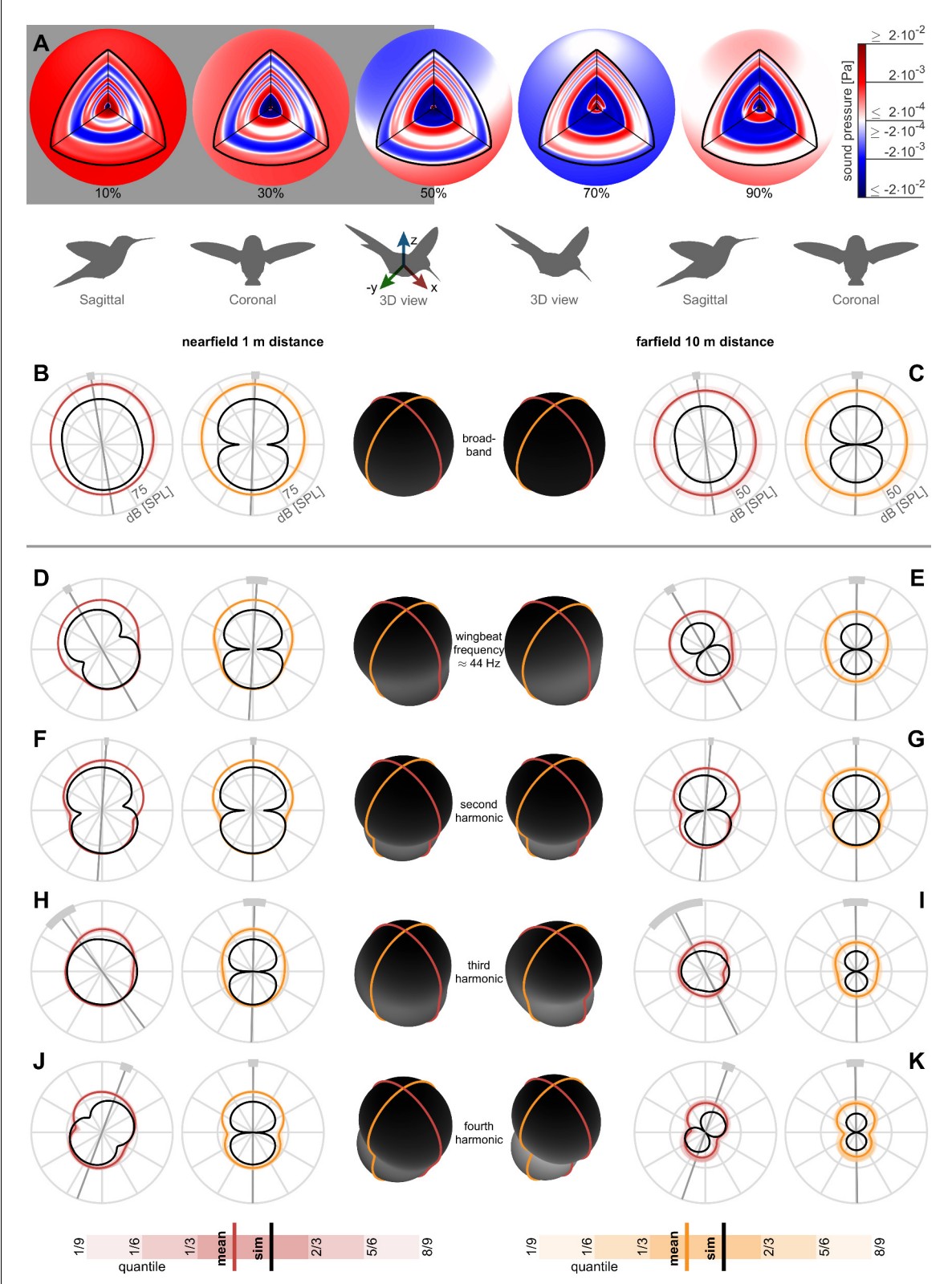

**Figure 3.** Nearfield versus farfield measured radial sound pressure level generated by a hovering hummingbird. (**A**) The full 3D broadband (from 3 to 500 Hz; animated in **Video 1**) pressure field measured over a wingbeat from bird #1 (oriented as the 3D view avatar) is shown across the spherical circumference at 1 m radius, the acoustic nearfield (outside the wing radius of the bird, 8 cm) and at 10 m radius, the acoustic farfield (wavelength of first wingbeat harmonic is 7.8 m). These 3D acoustic field reconstructions are based on the measurements from all arrays (**Figure 1D**). (**B**) At a nearfield

*Figure 3 continued on next page*

*Figure 3 continued*

distance of 1 m, the 3D broadband pressure surfaces can be represented with cross sections along the two key anatomical planes, the side/sagittal and front/coronal plane respectively, to visualize the broadband pressure directivity over the entire wingbeat. The mean pressure directivity trace for all birds is colored dark with color coding referring to the anatomical plane, the quantiles for each of the six birds are shaded light, and model prediction are shown in black. The overall pressure shape in 3D is plotted in the middle in black, which has a roughly spherical shape in the broadband holograms. (C) The 3D broadband pressure directivity at a farfield distance of 10 m. The waists of the individual lobes in each flight are smeared out due to small variations between the birds and their flights, obscuring the directivity in the average plots (individual traces shown in *Figure 3—figure supplement 1*). To show where the principle axes of the individual pressure lobes fall, we calculated the waistline pressure level between the minimum lobes and plot the directivity axis as the line perpendicular to the waistline (gray line, light gray arc ±1 SD; D, E). The broadband hologram can be further decomposed into contributions from the first harmonic. The measurement and simulations match better for the nearfield (computationally backpropagated) than for the farfield (computationally propagated). In the sagittal plane, the dipoles for both the measurement and model are tilted aft. This tilt can also be observed as a rotational mode associated with the wingbeat frequency in the longitudinal direction in the 3D animation for the first harmonic for bird #1 (*Video 2*). In contrast, the associated coronal dipoles are oriented vertical. The 3D pressure shape is also more oblong, as viewed by the ovoid black shape in the middle. (F, G) The sagittal and coronal dipoles of the second harmonic are oriented vertically in both the nearfield and farfield. This vertical orientation is associated with the vertical force generation occurring twice per wingbeat and is also visible in the 3D animation for the second harmonic (*Video 3*). (H, I) We observed a rotational mode in the 3D animation for the third harmonic (*Video 4*). (J, K) Both the sagittal and coronal dipoles of the fourth harmonic are oriented vertical in both the nearfield and farfield, which is also visible in the animation (*Video 5*). The third and fourth harmonic are decompositions of the first two modes; therefore, they share directivity similarities. Finally, the data driven model prediction in B-K (black contours) match the in vivo data reasonably well in amplitude considering the differences in peak spectrum amplitude noted in *Table 1*. There is also good agreement in the directivity of the predicted angles for the first two harmonics for both sagittal and coronal planes and for the first four harmonics for the coronal plane (*Table 2*), which matches the agreement in amplitude.

The online version of this article includes the following figure supplement(s) for figure 3:

**Figure supplement 1.** Individual directivity traces show waistlines.

determine how acoustic pressure scales with radial position (see Supplementary Information for details), which confirms $R_3$ is the appropriate radius. This acoustic radius agrees with wind turbine acoustics measurements at lower harmonics of the blade passing frequency (*Oerlemans et al., 2001*).

Starting at the time and location where the acoustic wave was generated by the unsteady force on the left and right wing, we numerically solve how the acoustic wave propagates outward into space to the location where we observe it with a microphone. Mathematically, the acoustic equation describes how the unsteady aerodynamic point force, $\boldsymbol{F}_{\mathrm{wing}}$, generated by the flapping wing generates an air pressure fluctuation, $p$, in the stationary atmosphere at the so-called 'retarded time', $t$, which radiates outward as a wave at the speed of sound, $a_o$, as follows *Lowson, 1965*:

$$
p = \underbrace{\left[ \frac{1}{4\pi |\boldsymbol{r}|^2 (1-M_r)^2} \left( \frac{1}{|\boldsymbol{r}|} \frac{(1-M^2)}{(1-M_r)} \left( \boldsymbol{r} \cdot \boldsymbol{F}_{\mathrm{wing}} \right) - \left( \boldsymbol{F}_{\mathrm{wing}} \cdot \boldsymbol{M} \right) \right) \right]}_{\text{nearfield}}
$$
$$
+ \underbrace{\left[ \frac{1}{4\pi a_o |\boldsymbol{r}|^2 (1-M_r)^2} \left( \left( \boldsymbol{r} \cdot \frac{\partial \boldsymbol{F}_{\mathrm{wing}}}{\partial t} \right) + \frac{1}{1-M_r} \frac{\partial M_r}{\partial t} \left( \boldsymbol{r} \cdot \boldsymbol{F}_{\mathrm{wing}} \right) \right) \right]}_{\text{farfield}} .
$$

(1)

The brackets indicate that the propagating pressure values, $p$, are evaluated at the retarded time, $t$. The vectorial distance from the moving point source on the flapping wing to the stationary microphone is measured by the vector, $\boldsymbol{r}$, in a Cartesian reference frame fixed to earth. The wing's velocity at the radial position where the point force acts, $v_{R3}$, is nondimensionalized with the acoustic wave velocity, $a_o$, the speed of sound, which defines in the Mach vector $\boldsymbol{M} \stackrel{\text{def}}{=} \boldsymbol{v_{R3}}/\boldsymbol{a_o}$. The Mach number is simply the magnitude of the Mach vector $M \stackrel{\text{def}}{=} |\boldsymbol{M}|$. Similarly, the convective Mach number, $M_r \stackrel{\text{def}}{=} \boldsymbol{M} \cdot \boldsymbol{r}/|\boldsymbol{r}|$, is simply the component of the Mach vector, $\boldsymbol{M}$, along the vector, $\boldsymbol{r}$, that runs from the wing source to the microphone. The acoustic pressure fluctuation, $p$, consists out of two components of which the respective strengths depend on how far the microphone is located away from the wing—measured in wavelengths of the acoustic frequency of interest (*Howe, 2014*). For a flapping hummingbird wing we choose the wingbeat frequency, because it is associated with the first harmonic we observe in the humming spectrum (*Figure 1F*), $\lambda_1 = a_0/f_1 \approx 343/44.2 = 7.8\ m$. The first term

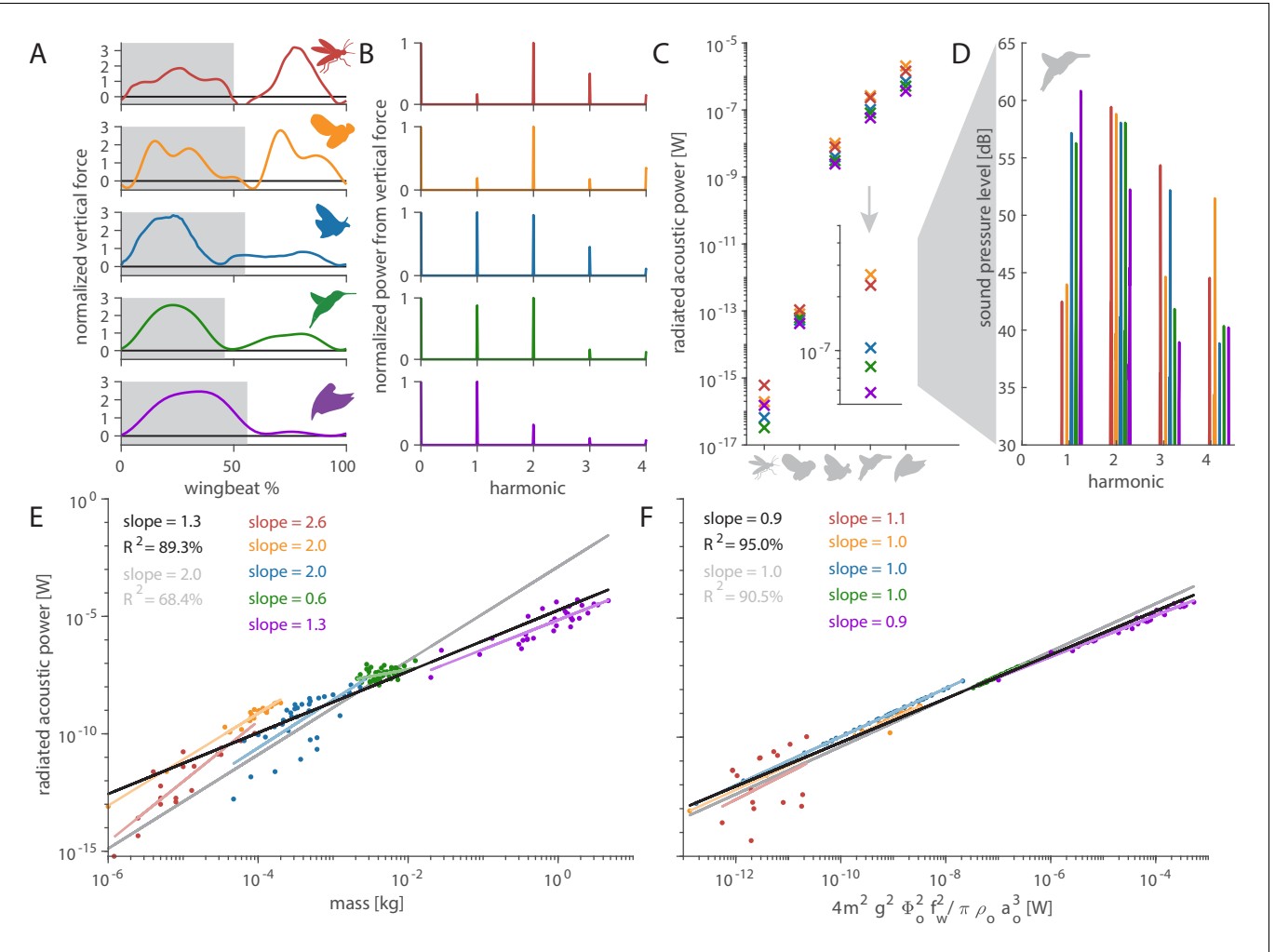

**Figure 4.** Distinct aerodynamic weight support profiles and non-allometric flapping wing scaling differentiates the acoustic spectrum and radiated power of flapping wing hum. (**A**) Representative aerodynamic weight support profiles of paradigm animals representing elongated flies, compact flies, butterflies and moths, hummingbirds, and generalist birds. The representative weight support profile was used to simulate the hum across animals in each group, with body mass varying over seven orders of magnitude and flapping frequency over three orders of magnitude. (**B**) The frequency content of these weight support profiles is distinct. Elongated flies and compact flies concentrate energy at the second harmonic and have substantial frequency content at higher harmonics compared to hummingbirds and hawkmoths, which have high first and second harmonics. In contrast, parrotlets concentrate most of their energy at the first harmonic. (**C**) Using our aeroacoustics model, we prescribed each of the five animals (gray avatars) all five weight support profiles (red, orange, blue, green, and purple datapoints match avatars in A) to determine how this affected the total radiated acoustic power of the wing hum (e.g. a fly was prescribed the respective weight support profiles of a mosquito, fly, hawkmoth, hummingbird, and parrotlet). The weight support profiles of the mosquito and fly consistently generate more radiated power than the profiles of the other animals. Differences between the paradigm animal groups across the different scales are primarily governed by nonlinear interactions between the acoustic parameters. The inset zooms in on the model results at hummingbird scale, which reveals the marked influence of weight support profile on radiated power over one order of magnitude. (**D**) At the hummingbird scale, the weight support profiles (**A and B**) differentiate between the overall decibel level and distribution across the first four harmonics (to enhance readability we slightly shifted each spectrum from the harmonic to the left). (**E**) We find these effects across the seven orders of magnitude across which body mass ranges for the 170 flying animals that perform flapping flight. The model is based on body mass, wing length, and flapping frequency of each individual species combined with the weight support profile of the associated paradigm animal (**A**). The computational results across all species (black line, best-fit scaling across all groups) show the simplified scaling law derived from the acoustic equations used in the model (gray line, predicted scaling result) closely matches the computational outcome for moths and butterflies (blue line). Other groups deviate appreciably from the acoustic scaling law prediction (colored lines, best-fit scaling per group), because their wing length and flapping frequency scale allometrically with body mass. (**F**) To test if the acoustic scaling law is reasonably accurate for all groups when allometric scaling is incorporated, we plot the simulated radiated acoustic power versus the scaling law: the product of force, stroke amplitude and flapping frequency squared (divided by the constant product of air density and speed of sound). On average this shows good agreement between the computational model (black line) and scaling law prediction (gray line) across all groups.

The online version of this article includes the following figure supplement(s) for figure 4:

*Figure 4 continued on next page*

*Figure 4 continued*

**Figure supplement 1.** The multiplicative factor $Rf_w/a_o \leq 0.01$ is much smaller than one for all 170 animals.
**Figure supplement 2.** The flapping Mach number $M_f \leq 0.1$ is small for all 170 animals.

in *Equation 1* dominates in the nearfield close to the wing up to a wavelength away from it. The associated pressure wave has a 3D dipole shape radiating in two opposing directions. Its strength is proportional to the force vector reorientation in space with respect to the radial vector, $r$, pointing from the source to the microphone. The second term dominates in the farfield starting at a wavelength away from the wing. The associated pressure wave has a 3D quadrupole shape along four primary directions. Its strength is proportional to the point force unsteadiness and the radial acceleration of its position in space. In the case of a hummingbird, the nearfield term decays exponentially with distance. This is because the hummingbird acts as a compact acoustic source (*Rienstra and Hirschberg, 2004*), since the wavelength at the wingbeat frequency (first harmonic) is much larger than the radius of the wing, $R$, the representative acoustic source length scale: $R/\lambda_1 = 0.007$ for $R = 0.058 \pm 0.003$ m. Consequently, a hummingbird wing acts as an approximate compact acoustic source up to its tenth wingbeat harmonic ($10 \cdot f_1$) with wavelength $\lambda_{10} = a_0/f_{10} \approx 343/442 = 0.78$ m. Because the hummingbird wing is acoustically compact across all the humming frequencies we study here, the wing is effectively acoustically transparent. The sound scattering over the wing is negligible and the time differences between local sound generating sources distributed over the wing can be ignored. Indeed, we observe a median difference of 0.1 dB between a single source model and a distributed model with 10 sources (*Figure 2—figure supplement 1A*, *Supplementary file 2*). The associated acoustic holograms of both models match spatially (*Figure 2—figure supplement 1B*), confirming hummingbird wings are compact acoustic sources at humming frequencies.

Using *Equation 1*, we calculate the resulting pressure fluctuation at each of the 2176 microphones in our acoustic arena to directly compare the simulated and measured humming sound up to the tenth harmonic. Beyond the tenth harmonic the ambient noise floor of the experiment is approached (*Figures 1F* and *2A*). Since the in vivo flapping frequency is used as an input to our model, *Equation 1*, there is exact frequency agreement between the modeled and in vivo spectra (*Figure 2A*). Spatially, the model captures the wingstroke transitions in the top and front arrays in the holograms (*Figure 2B*). The model and recordings are in good agreement, because the difference in the magnitude of the sound pressure is ~4 dB or less for the first four harmonics (maximum difference between the model and the measurement ±1 SD; *Figure 2A*, *Table 1*). The first four harmonics represent most of the radiated harmonic power: ~99% of the simulated power and ~67% of the measured power for ±2.5 Hz bands around each wingbeat harmonic up to 180 Hz. The percentage difference is due to at least three factors: (i) harmonics beyond the fourth contribute more power in the measured spectrum than in the simulated spectrum (*Table 2*), (ii) the experiment's ambient noise floor is substantially higher than the computational noise floor (*Figure 1F*), and (iii) some low amplitude tonal noise sources observed between harmonics cannot be attributed to humming (*Figure 2A*). The differences across all 10 harmonics may include some acoustic scattering by the wing and body, possible wingtip flutter (*Sane and Jacobson, 2006*) and turbulent vortex dynamics contributions occurring multiple times during a wingbeat, so they overlap with the measured harmonics. The magnitude of these effects combined is bounded by the differences in the measured

**Table 1.** The measured and predicted sound pressure level peaks across the first 10 harmonics.
The measurement and model are close up to the fourth harmonic. The over-prediction for the seventh harmonic and up may be attributed to frequency mixing. Past the tenth harmonic, we approach the ambient noise floor for the measurements.

| Harmonic | 1st | 2nd | 3rd | 4th | 5th | 6th | 7th | 8th | 9th | 10th |
|---|---|---|---|---|---|---|---|---|---|---|
| Measurement [dB] ± SD [dB] | 60.8 ±1.2 | 60.0 ±1.2 | 47.9 ±2.6 | 46.4 ±3.4 | 42.7 ±3.2 | 41.7 ±3.2 | 34.0 ±2.6 | 28.8 ±3.7 | 23.6 ±3.1 | 25.2 ±2.3 |
| Model [dB] | 55.3 | 57.0 | 48.4 | 40.5 | 33.4 | 30.9 | 32.9 | 28.1 | 31.5 | 23.1 |

**Table 2.** The measured and predicted broadband pressure directivity angles match.
Aft tilt is evident in the sagittal planes, whereas the coronal planes show vertical directionality associated with vertical force generation. Harmonic modes 1–4 match well in the coronal plane and modes 1 and 2 match well in the sagittal plane.

| Broadband | Sag near | Cor near | Sag far | Cor far |
|---|---|---|---|---|
| Measurement [°] | 99.4 | 88.0 | 97.4 | 89.1 |
| ± SD [°] | ±3.1 | ±3.4 | ±3.2 | ±4.6 |
| Model [°] | 102.3 | 90.2 | 97.8 | 90.0 |
| Sagittal nearfield | 1st | 2nd | 3rd | 4th |
| Measurement [°] | 119.7 | 86.3 | 126.9 | 69.6 |
| ± SD [°] | ±3.4 | ±1.9 | ±13.7 | ±5.0 |
| Model [°] | 125.8 | 99.8 | 82.7 | 44.3 |
| Sagittal Farfield | 1st | 2nd | 3rd | 4th |
| Measurement [°] | 120.4 | 85.4 | 116.6 | 70.8 |
| ± SD [°] | ±4.4 | ±2.2 | ±24.4 | ±5.1 |
| Model [°] | 125.6 | 99.8 | 78.9 | 44.6 |
| Coronal nearfield | 1st | 2nd | 3rd | 4th |
| Measurement [°] | 86.7 | 89.7 | 88.7 | 89.8 |
| ± SD [°] | ±8.8 | ±2.7 | ±9.3 | ±4.1 |
| Model [°] | 89.9 | 90.2 | 89.9 | 90.0 |
| Coronal Farfield | 1st | 2nd | 3rd | 4th |
| Measurement [°] | 89.4 | 90.2 | 90.5 | 90.9 |
| ± SD [°] | ±6.6 | ±2.8 | ±10.3 | ±7.2 |
| Model [°] | 90.1 | 90.0 | 90.0 | 90.0 |

and simulated spectra (*Figure 2A*), which ranges from ~0.5 to ~7.0 dB (min. and max. difference ±1 SD; *Table 1*).

## Dipole acoustic directivity patterns align with gravitational and anatomical axes

The directivity of the acoustic pressure field varies between harmonics. Odd harmonics are associated with a rotational pressure fluctuation mode while even harmonics are associated with a vertical pressure fluctuation mode. To assess the near and farfield directivity, we reconstruct 3D broadband pressure fields (across 3–500 Hz) over an entire wingbeat during stationary hovering flight. The reconstructed pressure fields start out at a radius of 8 cm centered on the body such that the inner spherical surface encloses the hummingbird (the wing radius with respect to the body center is 5.8 ± 0.3 cm) and the outer spherical surface ends at a radius of 10 m (*Figure 3A*; animation in *Video 1*). To evaluate acoustic pressure directivity in the nearfield (1 m distance, ~8.6 wingspans, *Figure 3B*) and farfield (10 m distance, ~86 wingspans, *Figure 3C*), we calculate the cross-sections of the pressure field in the sagittal (side) and coronal (frontal) anatomical planes. Averaging directivity plots across all birds and flights, we find the 3D broadband pressure surface is roughly spherical in the nearfield and farfield (plotted in the middle of *Figure 3B,C* in black). To observe the contribution from each harmonic, we decompose the broadband pressure with a bandwidth of ±2.5 Hz around each of the first four harmonics (*Figure 3D–K*). Each individual directivity plots' principal axis is oriented perpendicular to the waistline of the dipole lobes we measured (average, gray line; ±1 standard deviation, light gray arc) and simulated (comparison in *Table 2*). The principal axis is mostly vertical because the net aerodynamic force generated during hover opposes gravity. The dipole shape also manifests in the ovoid 3D pressure surface at these harmonics (*Figure 3D–K*).

The orientation of the measured and predicted broadband holograms in the sagittal and coronal plane agrees within one standard deviation or less (*Figure 3B,C*; *Table 2*). This is explained by the reasonable correspondence between the measured and predicted directivity (*Figure 3D–G*) and

amplitude (*Figure 2A*) of the first and second harmonic, which have the largest amplitudes across all harmonics. Both the near and farfield broadband directivity plots are pointed aft in the sagittal plane because the dominant first harmonic is oriented aft. The correspondence between the predicted and measured amplitude (*Table 1*) and directivity in the sagittal (but not coronal) plane (*Table 2*) weakens starting at the fourth and third harmonic respectively. Higher harmonics contribute less to the broadband directivity, because their amplitude is much lower (<48 dB beyond the third harmonic, *Table 1*). Due to the symmetry between the left and right wing, the coronal directivity points upwards at 90° across all measured and simulated harmonics (*Figure 3*, *Table 2*), showing the hummingbirds performed symmetric hovering flight.

In summary, the first harmonic of the hummingbird hum is formed by an acoustic dipole, tilted aft in the coronal plane, which corresponds to the fluctuation of the net vertical and asymmetric horizontal force over a wingbeat. The associated rotational mode can be observed in the time-dependent 3D hologram (*Video 2*). The second harmonic is formed by an upward pointing dipole, corresponding to the vertical force generation that occurs twice per wingbeat (*Figure 3F,G*). This is visible as a vertically oriented mode in the time-dependent 3D hologram (*Video 3*). The third harmonic consists also of a rotational mode like the first harmonic (*Figure 3H,I*), as seen in the time-dependent 3D hologram (*Video 4*). Likewise, the fourth harmonic consists of a vertical mode like the second harmonic (*Figure 3J,K*; *Video 5*).

## Extension of the acoustic model across animals that flap their wings

Using our model, we predict the acoustic sound generated by flapping wings for a wide range of insects and birds that hover or perform slow flapping flight during takeoff and landing across seven orders of magnitude in body mass, $m$, and three orders of magnitude in wing flapping frequency, $f_w$. We generalize the flapping animals we consider here into five distinct groups for which we found data: generalist birds (*Aves* except *Trochilidae*), hummingbirds (*Trochilidae*), moths and butterflies (*Lepidoptera*), compact flies (*Cyclorrhapha*), and elongated flies (*Nematocera*), which fly with marked shallower stroke amplitudes than compact flies. Since 3D aerodynamic force and wing kinematics data are not available for all these species, and most of the radiated acoustic sound is directed vertically (*Figure 3B–K*), we simplified the model. We chose a well-studied animal for which a wingbeat-resolved vertically-oriented force component has been reported previously to act as a paradigm for each group. Respectively, the vertical force of pacific parrotlets (*Forpus coelestis Chin and Lentink, 2017*) for generalist birds, the vertical force of Anna's hummingbird (*Calypte anna*; *Ingersoll and Lentink, 2018*) for hummingbirds, the lift force of hawkmoths (*Manduca sexta Zheng et al., 2013*) for moths and butterflies, the lift force of mosquitos (*Culex quinquefasciatus*; *Bomphrey et al., 2017*) for elongated flies and the net force of *Drosophilid* flies (*Drosophila hydei*; *Muijres et al., 2014*) for compact flies (*Supplementary file 5*). To simplify the comparison further, we approximate the stroke plane as horizontal and the normalized lift profile to have the same shape as the reported vertically oriented force profile, so that the lift generated during a wingbeat sums up to body weight for all associated species in the same way. To calculate the associated drag profile, we used previously reported quasi-steady lift/drag ratio data for Anna's hummingbirds (*Ingersoll and Lentink, 2018*; *Kruyt et al., 2014*) and assume it is representative for all animals. Finally, to compute the acoustic field for each animal's wing, we locate the lift and drag force at the third moment of area of a hummingbird wing, 55% of the wing radius (which compares to 58% for parrotlets *Chin and Lentink, 2017*). In our comparison, we make the exact same approximations for hummingbirds as we do for the other animals. Despite these assumptions, the simplified model matches the original model for a hummingbird well (*Figure 2—figure supplement 2*, *Supplementary file 3*). Between each of the four groups, the instantaneous weight support, stroke amplitude, and frequency content throughout the wingbeat change based on the associated paradigm animal (*Figure 4A,B*). In contrast, the mass, wingspan, and flapping frequency change across all individual animals in each group. Calculating the ratio of the wing length versus acoustic wavelength at the wingbeat frequency across all species, we find $Rf_w/a_o = R/\lambda_1 \leq 0.01$ (*Figure 4—figure supplement 1*). Indeed, synchronized acoustic and video recordings show that the measured first acoustic harmonic overlaps with the wingbeat frequency across insects (*Cator et al., 2009*; *Aldersley and Cator, 2019*) and hummingbirds (*Figure 1*) as well as other birds and bats (*Boonman et al., 2020*). Thus, similar to the hummingbird, the flapping wings of all these animals act as compact acoustic sources from the first to tenth harmonic. Furthermore, because wing length is inversely proportional to flapping frequency

(*Greenewalt, 1962*), the assumption of acoustic compactness holds across species. Consequently, the humming sound generated across flapping animal wings can be modeled accurately with a single point force source per wing half, similar to what we found for hummingbirds (*Figure 2—figure supplement 1*). This even holds for mosquito buzz, the most extreme case among our five paradigm animals, because the mosquito wing's compactness, $R/\lambda_1 = 0.006$, is equivalent to that of a hummingbird's 0.007.

The weight support profiles of each of the five paradigm animals has distinct harmonic content (*Figure 4B*). To understand how this drives acoustic power and timbre, we use our acoustic model to assign each of the five paradigm animals all five weight support profiles. For example, we variously assign the weight support profile of a mosquito, fly, hawkmoth, hummingbird, and parrotlet to our hummingbird model. This allows us to investigate the weight support profile's effects on differences in radiated acoustic power (*Figure 4C*) and the acoustic spectrum (*Figure 4D*). The weight support profiles of the mosquito and fly consistently generate more acoustic power and sound pressure than the other weight support profiles. Lastly, we extend the acoustic model from the five paradigm animals to 170 animals across the five groups. Body mass and flapping frequency for hummingbirds, compact flies, elongated flies, and moths and butterflies were obtained from *Greenewalt, 1962*, while the values for larger birds were obtained from *Pennycuick, 1990*; *Figure 4E,F*. Comparing the model simulation results with the isometric scaling relation we derived based on the model (*Equations A25–50*) shows that radiated acoustic power scales allometrically with body mass (*Figure 4E*) except for compact flies and moths and butterflies, which scale isometrically. Considering flapping wing parameters are known to scale allometrically with body mass, we test the scaling law itself (*Figure 4F*), which collapses the data well on average across species (average slope = 0.9; ideal slope = 1), confirming the scaling law represents our model.

## Discussion

### Oscillating lift and drag forces explain wing hum timbre

Our idealized aeroacoustic model shows the hummingbird's hum originates from the oscillating lift and drag forces generated by their flapping wings. Remarkably, the low frequency content in the aerodynamic forces also drives higher frequency harmonics in the acoustic spectrum of the wing hum. The higher harmonics originate from nonlinear frequency mixing in the aeroacoustic pressure equation between the frequency content in the wing's aerodynamic forces and kinematics. The predicted humming harmonics of the wingbeat frequency overlap with the measured acoustic spectrum (averaged over all microphones). In addition to the good frequency match, the sound pressure level magnitudes of the first four harmonics match with a difference of 0.5–6.0 dB (*Table 1*). This agreement is similar or better compared to more detailed aeroacoustic models of drone and wind turbine rotors, that predict noise due to blade-wake interactions and boundary layer turbulence (*Oerlemans and Schepers, 2009*; *Zhang et al., 2018*; *Wang et al., 2019*). Further, comparing the measured and predicted spatial acoustic-pressure holograms for the top and front arrays (reconstructed *holograms* at a plane 8 cm from the bird; *Figure 2B*), we find that the hologram phase, shape, and magnitude correspond throughout the stroke. The regions of high and low pressure in the hologram are associated with wing stroke reversals, similar to the pressure extrema observed at stroke reversal in computational fluid dynamics simulations of flapping insect wings (*Geng et al., 2017*; *Seo et al., 2019*; *Nedunchezian et al., 2018*).

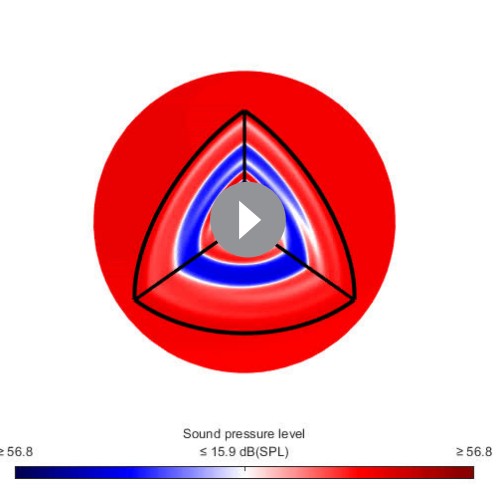

Sound pressure level
≥ 56.8 ≤ 15.9 dB(SPL) ≥ 56.8

**Video 1.** The 3D broadband hologram shows how pressure waves emanate from the nearfield to farfield.
https://elifesciences.org/articles/63107#video1

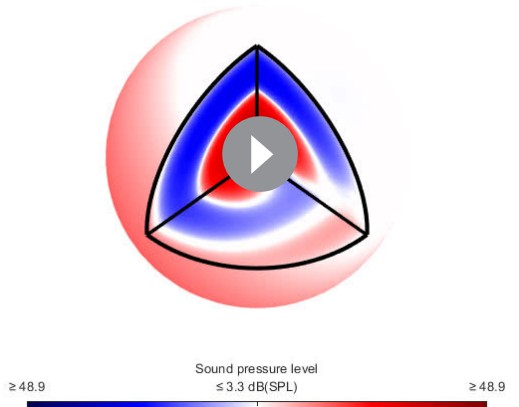

Sound pressure level
≥48.9    ≤3.3 dB(SPL)    ≥48.9

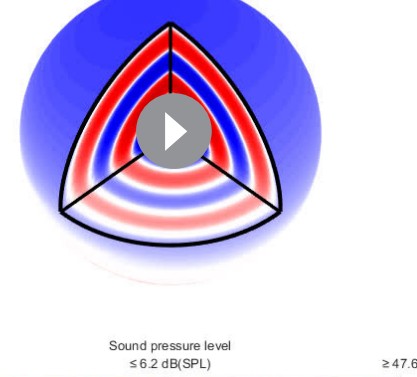

Sound pressure level
≥47.6    ≤6.2 dB(SPL)    ≥47.6

**Video 2.** The 3D hologram for the first harmonic conveys the rotational mode associated with the tilted dipole.
https://elifesciences.org/articles/63107#video2

**Video 3.** The 3D hologram for the second harmonic conveys the vertical mode associated with the vertically oriented dipole.
https://elifesciences.org/articles/63107#video3

Even though the input forces were lowpass filtered beyond the fourth harmonic, the amplitudes of higher harmonics are predicted. This is due to two distinct stages of nonlinear frequency mixing in our wing hum model: (*i*) the calculation of the resulting aerodynamic force vector generated by each flapping wing and its oscillatory trajectory in space, and (*ii*) the calculation of the resulting acoustic pressure waves (see Supplementary Information for details).

Our acoustic model predicts hum harmonics that lie in an intermediate frequency range between the wingbeat frequency (~40 Hz) and the lower bound of feather sonations (typically >300 Hz; *Clark et al., 2013a*; *Clark et al., 2013b*). Hence our model allows for an objective contrast between wing hum sound and other possible aerodynamic noise generation mechanisms. Indeed, we observe small tonal peaks between the prominent harmonics in *Figure 2A* that are not radiated by the oscillating aerodynamic forces generated by the flapping wing, according to our hum model. Consequently, these low amplitude peaks must radiate from another acoustic source such as aeroelastic feather flutter (*Clark et al., 2011*) or vortex dynamics (*Ellington et al., 1996*).

In the under-studied frequency regime of the hum, the first two harmonics are paired as they have similar sound pressure levels (*Figure 2A*). For the hummingbird, the pairing of the first and second harmonics is due to the dominance of the pressure differential generated twice per wingbeat during the downstroke and upstroke. The associated substantial weight support during the upstroke (*Figure 1B*; *Ingersoll and Lentink, 2018*) has been found across hummingbird species (*Ingersoll et al., 2018*), which generalizes our findings. The sound pressure level pairing also mirrors the harmonic content in the lift and drag forces (*Figure 1C*) as well as the stroke and deviation kinematics (*Figure 1I*). Given that the first and second harmonics dominate both the forces and kinematics spectra, the harmonic content of the resulting acoustics is a mixture of these two. The third harmonic and beyond resemble the first paired harmonic because they are associated with the noise generation mechanisms of the first two harmonics (*Rienstra and Hirschberg, 2004*). In concert, the first four harmonics constitute most of the acoustic radiated power of the hum timbre—the distinct sound quality that differentiates sounds from distinct types of sources even at the same pitch and volume—which is determined by the number and relative prominence of the higher harmonics present in a continuous acoustic wave (*Sethares, 2005*).

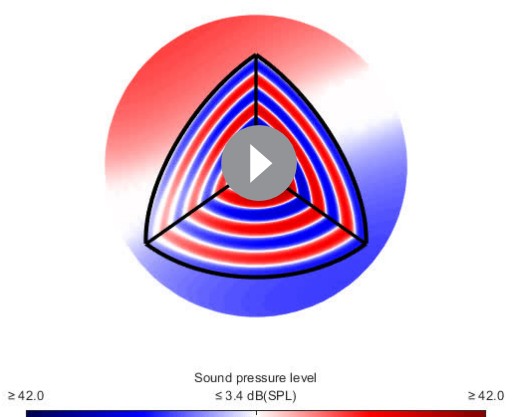

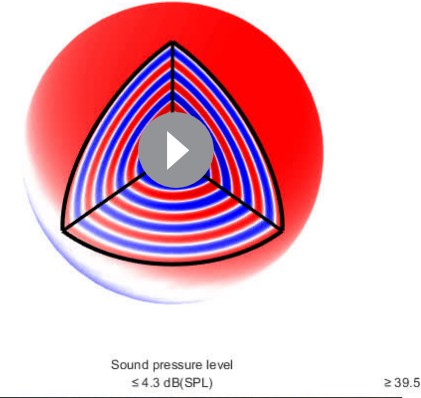

Sound pressure level
≥ 42.0        ≤ 3.4 dB(SPL)        ≥ 42.0

Sound pressure level
≥ 39.5        ≤ 4.3 dB(SPL)        ≥ 39.5

**Video 4.** The 3D hologram for the third harmonic conveys the rotational mode associated with the tilted dipole.
https://elifesciences.org/articles/63107#video4

**Video 5.** The 3D hologram for the fourth harmonic conveys the vertical mode associated with the vertically oriented dipole.
https://elifesciences.org/articles/63107#video5

## Wing hum acoustic directivity and orientation depends on harmonic parity

Acoustic directivity is consistent from near to farfield, but changes based on the harmonic. In the 3D holograms, the dipole structures are associated with the high vertical forces to offset weight (*Ingersoll and Lentink, 2018*). These dipole orientations are not evident in the broadband holograms (*Figure 3B,C*) because slight variations between the flights are averaged and smear out the dominant dipole lobes (individual flights for each directivity plot shown in *Figure 3—figure supplement 1*). The first and third harmonics resemble dipoles that are tilted aft. For example, for the first harmonic in the sagittal plane in both the nearfield and farfield, the dipole is tilted aft (*Figure 3D,E*; *Table 2*), which is associated with the pressure generated during the downstroke once per wingbeat. In contrast, second and fourth harmonics are more vertically oriented. The second harmonic is directed upwards in the nearfield and farfield (*Figure 3F,G*; *Table 2*) and is associated with the pressure generation for the vertical weight support that occurs twice per wingbeat. The third and fourth harmonics have more complex shapes (*Figure 3H–K*) that bear resemblances to the first two because they are associated with the first two harmonics (*Rienstra and Hirschberg, 2004*). The acoustic model also shows these directionality effects over the first two harmonics in the sagittal and coronal near and farfield. In contrast, the simulation has more symmetry between the upstroke and downstroke, resulting in a symmetric and better-defined dipole structure. The dipoles that we measured for the first four hummingbird harmonics (*Figure 3D–K*) are strikingly similar to the ones found for hovering insects in computational fluid dynamics simulations (*Geng et al., 2017*; *Seo et al., 2019*). Although the mosquito dipoles are oriented more horizontally, because their wings generate unusually high drag at these harmonics (*Seo et al., 2019*), due to their particularly shallow wingstroke (*Bomphrey et al., 2017*).

## Acoustic model explains perceived hum loudness and timbre of birds and insects

The sound magnitude that flapping wings produce depends heavily on the weight the flapping wings must support, and the timbre depends on the unique frequency content of each weight support profile (*Figure 4B*). Flies and mosquitos are orders of magnitude lighter than our three other paradigm animals and produce less acoustic power accordingly (*Figure 4C*). Yet the fly and mosquito weight support profiles have the highest harmonic content (*Figure 4B*) and therefore, when all else is equal, consistently radiate the most power (*Figure 4C*). In contrast, the parrotlet weight

support profile has the lowest harmonic content (*Figure 4B*); with most of the force being generated once per wingbeat during the downstroke, hence it radiates the least power when all else is equal (*Figure 4C*). For hummingbirds and hawkmoths, the proportion of weight support in upstroke versus downstroke is similar (*Geng et al., 2017*; *Ingersoll and Lentink, 2018*); this gives them roughly similar vertical force profiles and leads to similar acoustic power (*Figure 4C*). The effect of altering the weight support profile is also visible in the acoustic spectrum. At the scale of a hummingbird (*Figure 4C*, inset), the prescribed weight support profiles distinguish the distribution of the overall decibel level for the first four harmonics (*Figure 4D*). This explains why flies and mosquitos may seem loud relative to their small size: while they have little mass, it is partially offset by the high harmonics in their weight support profiles. Furthermore, it is the higher harmonics present in the weight support profile that directly affect the perceived quality of the sound—the timbre.

## Radiated acoustic power scales allometrically in birds and elongated flies

Body mass is a strong predictor of radiated acoustic power because the aerodynamic forces needed to sustain slow hovering flight must be proportionally larger for heavier animals (*Weis-Fogh, 1973*; *Altshuler et al., 2010*; *Skandalis et al., 2017*). The associated increase in aerodynamic force amplitude drives acoustic pressure (*Equation 1*). The resulting radiated acoustic power, $P$, scales with the square of the acoustic pressure, $p$ (*Equation A25*). Increasing flapping frequency also increases the radiated acoustic power; flapping faster requires more power from the animal and injects more acoustic energy into the air. Applying scaling analysis to *Equation 1* (derived in Supplementary Information; *Equations A25–50*), we can predict the order of magnitude of the radiated acoustic power in the farfield (*Howe, 2014*):

$$P_o = \frac{4F_o^2\Phi_o^2 f_w^2}{\pi\rho_o a_o^3} \approx 2.5 \cdot 10^{-6}\Phi_o^2 m^2 f_w^2, \tag{2}$$

where the subscript '$o$' corresponds to the reference value and $F_0 = mg$ is the aerodynamic force magnitude required to maintain hover. The resulting acoustic power law scales with the product of wing stroke amplitude, $\Phi_o$, body mass, $m$, and wingbeat frequency, $f_w$, squared. Further, since $\Phi_o$ is dimensionless, it has order of magnitude one, measured in radians, across flapping birds (*Nudds et al., 2004*) and insects (*Azuma, 2006*). The remaining terms, $4/\pi$, the gravitational constant $g = 9.81\mathrm{ms}^{-2}$, the air density $\rho_o \approx 1.23\mathrm{kgm}^{-3}$, and speed of sound in air, $a_o \approx 343\mathrm{ms}^{-1}$ are constants that determine the factor $2.5 \times 10^{-6} \mathrm{kg}^{-1}\mathrm{s}^{-1}$ between the radiated acoustic power and its scaling variables.

When acoustic power is plotted as a function of mass (*Figure 4E*), the predicted exponent of 2.0 is higher than the observed average exponent of 1.3. Among the five groups, compact flies and moths and butterflies do match the scaling law prediction, showing their acoustic power scales isometrically with body mass. The other groups scale allometrically with either higher, elongated flies, or lower, hummingbirds and other birds, exponents of body mass. Allometric divergence can more readily explain why larger hummingbirds are quieter, because they have disproportionally larger wings combined with an approximately constant wing velocity across an order of magnitude variation in body mass, which is thought to maintain constant burst flight capacity (*Skandalis et al., 2017*). Conversely, for insects, the gracile bodies and larger wings of moths and butterflies are offset by the higher flapping frequency of compact flies. Therefore, flies use asynchronous flight muscles to achieve these high flapping frequencies (*Deakin, 1970*). Large, elongated flies are unusually noisy for their body mass, with radiated acoustic power values well above the average scaling law (*Figure 4E*). The disproportional noise generated by elongated flies is due to two combined effects: the higher harmonic content of their weight support profile (*Figure 4A,B*) and their consistent allometric acoustic power scaling (*Figure 4E*).

The difference between the scaling exponents for mass is primarily due to allometric scaling of wingbeat frequency with body mass because the simulated acoustic power scales with the right-hand side of scaling *Equation 2* with an exponent of 0.9 (on average), close to 1 (*Figure 4F*). Scaling *Equation 2* is precise for birds, compact flies, and moths and butterflies, but the two other groups scale allometrically: larger birds get more silent (slope = 0.9) while elongated flies (1.1) get louder than predicted by isometric scaling incorporating the allometric body mass and wing frequency

relationship. The deviation may be partially explained by variation in wing stroke amplitude (*Bomphrey et al., 2017*; *Nudds et al., 2004*; *Azuma, 2006*). Further, body size and wingspan in insects are highly variable amongst individuals of even the same species (*Debat et al., 2003*), which may explain the larger variation. Finally, the assumptions underpinning our scaling analysis may explain some deviation.

## New tool to interpret complex bioacoustics behavior

The extension of our simple model to predict flapping wing hum across a wide range of species (*Figure 4*) makes it a useful tool to study insects, birds, and bats performing a variety of complex behaviors. Like the acoustic power scaling law (*Equation 2*), *Equation 1* can be simplified further (*Figure 4—figure supplements 1,2*) for comparative biomechanical and neuroethological studies:

$$p = \underbrace{\frac{1}{4\pi|\boldsymbol{r}|^3}\left(\boldsymbol{r}\cdot\boldsymbol{F}_{\text{wing}}\right)}_{\text{nearfield}} + \underbrace{\frac{1}{4\pi a_o|\boldsymbol{r}|^2}\left(\boldsymbol{r}\cdot\frac{\partial\boldsymbol{F}_{\text{wing}}}{\partial t}+\frac{4\Phi_o R f_w^2}{a_o}\left(\boldsymbol{r}\cdot\boldsymbol{F}_{\text{wing}}\right)\right)}_{\text{farfield}}. \tag{3}$$

Our study shows how this model can elucidate the mechanistic origin of wing hum timbre (and modulation) in vivo by integrating acoustic recordings with high-speed videography and aerodynamic force recordings. Likewise, we showed it can be used to make predictions or interpret acoustic measurements by integrating a simplified wing kinematics and aerodynamic force model. It can also be used to estimate the auditory detection distance of wing hum by combining it with an audiogram. Finally, the ability to distinguish between the nearfield *versus* farfield provides an additional lens for behavioral inquiry.

The predicted range over which wing hum can be perceived is even larger in rock pigeons; approximately four meters or ~12 wing radii (*Columba livia*: flapping frequency 7 Hz, mass 400 g, wing length 32 cm; *Pennycuick, 1968*). The perception distance scales up with body mass (*Figure 4E*) and the auditory threshold of pigeons is exquisitely sensitive to the wingbeat frequency (*Kreithen and Quine, 1979*), which can thus potentially inform flocking behavior (*Larsson, 2012*). Conversely, while the low-frequency oscillating aerodynamic forces also radiate high-frequency humming harmonics up to the tenth wingbeat harmonic (*Figure 2A*, *Table 1*) and beyond, the corresponding decibel amplitudes are insignificant compared to harmonics close to the wingbeat frequency (*Figure 1F*). This helps explain why some birds rely on specialized flight feathers that sonate loudly at high frequency to signal over longer distances how they are flapping their wings during flock takeoff (*Hingee and Magrath, 2009*; *Niese and Tobalske, 2016*; *Murray et al., 2017*), mating displays (*Clark et al., 2016*) and displays to defend courting territories (*Miller and Inouye, 1983*). Perception of wing whoosh also has implications for bird-insect predation, because moths have been shown to respond to the wingbeat hum of birds in playback experiments (*Fournier et al., 2013*).

Finally, an acoustic model analogous to the one we present here has recently been used to simulate mosquito buzz (*Seo et al., 2019*) in conjunction with computational fluid dynamics to predict how aerodynamic forces (*Bomphrey et al., 2017*) color the mosquito's aerial courtship song (*Cator et al., 2009*). Intriguingly, whereas mosquitos fly with a shallow wing stroke to generate high harmonic content, fruit flies do not (*Bomphrey et al., 2017*). When fruit flies use their wing as an aeroacoustic instrument during terrestrial courtship serenades; however, they reduce their stroke amplitude to a similar degree (*von Schilcher, 1976*; *Bennet-Clark and Ewing, 1968*), which likely colors their timbre as in mosquitos (*Figure 4A–D*).

## Conclusion

Our acoustic model explains how the oscillating lift and drag forces generated by each wing of a hovering hummingbird radiate the distinctive humming timbre. It integrates in vivo 3D aerodynamic force and wing kinematics measurements and is corroborated spatially and temporally through in vivo nearfield acoustic holography. The measurements and model show that hovering hummingbirds generate a highly directional hum. The broadband acoustic pressure is primarily oriented downward opposing gravity, while the acoustic directivity and orientation of the harmonic components depend on harmonic parity. The model explains how perceived differences in hum loudness and timbre across birds and insects stem primarily from the harmonic content in the aerodynamic weight

support profile. Higher harmonic content throughout the wing stroke makes flies and mosquitos buzz, equivalent first and second harmonic content makes hummingbirds hum, while dominant first harmonic content gives birds their softer whoosh. The associated scaling relation for radiated acoustic power shows how it is proportional to the product of stroke amplitude, body mass and wingbeat frequency squared. Our scaling analysis across 170 different animals in slow hovering flight reveals how the radiated acoustic power scaled with mass. Allometric deviation explains why larger birds radiate less acoustic power than expected and why elongated flies have a remarkably loud buzz as perceived by a casual observer. Finally, our acoustic model and scaling equation can help neuroethologists and bioacousticians interpret the loudness and timbre of the hum generated by flapping winged animals performing complex behaviors as well as guide bioinspired engineers how to design more silent flapping robots (*Wood, 2008*; *Keennon et al., 2012*).

## Materials and methods

### 3D aerodynamic force platform setup

The 3D aerodynamic force platform flight arena consisted of a 0.5 × 0.5 × 0.5 m (height, width, depth) chamber, where each of the six sides is an instrumented carbon fiber force plate that mechanically integrates pressure and shear forces generated by the freely flying hummingbird (*Ingersoll and Lentink, 2018*; *Lentink et al., 2015*; *Hightower et al., 2017*). Three high-speed stereo cameras captured the wingbeat kinematics through three orthogonal imaging windows in the plates. Each plate is statically determined and attached to three vee blocks (VB-375-SM, Bal-tec), each instrumented by a Nano 43 6-axis force/torque sensor (4000 Hz sampling rate, lowpass filtered with an eighth order digital lowpass Butterworth filter at 180 Hz, silicon strain gage based, with SI-9–0.125 calibration, 2 mN resolution, ATI Industrial Automation). There are also two force sensors instrumenting a beam attached to the artificial flower to measure hummingbird contact forces and body weight. For detailed analysis, we selected 3D force traces over five consecutive wingbeats per flight ($N$ = 6 birds, each bird did two flights, $n$ = 5 wingbeats per flight for 60 wingbeats total) for which we manually tracked the 3D wing kinematics of four points on the bird (right shoulder, distal end of the leading-edge covert, wingtip, and tip of the fifth primary feather). We recorded wingbeat kinematics through three orthogonal acrylic access ports using stereo high-speed videography at 2000 Hz using three pairs of DLT calibrated (*Hedrick, 2008*) cameras (four Phantom Micro M310s, one R-311, and one LC310; Vision Research). We filtered the kinematics with a fourth order digital lowpass Butterworth filter with a cutoff frequency of 400 Hz (~10 times the wingbeat frequency).

### Acoustic microphone array setup and holographic analysis

The acoustic setup consisted of a chamber that is 0.3 × 0.9 ×0.9 m (height, width, depth). The sides of the chamber were made of IR transparent acrylic (Plexiglass 3143) to allow visual access into the chamber while controlling what the hummingbird views from inside the chamber. Two battery-powered LED lights (Neewer CN126) sustained a constant light level of 3000 lux at the flower. Combined, the arrays surrounded the hummingbird with 2176 microphones (of which 25 ± 7 were disabled during each measurement; see Supplementary Information for details) while it freely hovered in front of a flower to feed. The top and bottom arrays (Sorama CAM1Ks) each consist of 1024 microelectromechanical (MEMS) microphones, while the two frontal arrays (Sorama CAM64s) feature 64 microphones each with a sampling frequency of 46,875 Hz. During the actual flight, these arrays were covered by an acoustically transparent cloth (Acoustone speaker grille cloth) to protect both the bird and the microphones. To limit wall effects encountered in flight arenas (*Hightower et al., 2017*), the feeder was centered 15 cm horizontally from the edge and 15 cm above the bottom array. The sides of the acoustic arena featured optically accessible panels in the infrared range, which were used to film the hummingbirds with four direct linear transformation calibrated high-speed infrared cameras at 500 fps. The 3D pressure field was reconstructed from the planar array measurements using broadband nearfield acoustic holography (NAH). Each frequency component of the holograms was regularized independently using a Bayesian evidence method (*Wijnings, 2015*) before adding them all together to create the broadband NAH results. To reduce distortions due to frequency leakage, linear predictive border padding (*Scholte, 2008*; *van Dalen et al., 2012*) was

applied to the time signals. The radial directivity was computed using spherical NAH (*Williams, 1999*).

## Acknowledgements

We thank A Stowers and E Chang for help with the experiments, C Lawhon for fabricating the force plates, and JM Knapp for manuscript feedback. This work was supported by NSF Faculty Early Career Development (CAREER) Award 1552419 to DL. BJH. was supported by the NSF Graduate Research Fellowship and the Stanford Graduate Fellowship. PWAW. was supported by Netherlands Organisation for Scientific Research Program ZERO (P15-06). DDC. was supported by the National Defense Science and Engineering Graduate Fellowship and Stanford Graduate Fellowship.

## Additional information

### Competing interests

David Lentink: Reviewing editor, *eLife*. The other authors declare that no competing interests exist.

### Funding

| Funder | Grant reference number | Author |
| --- | --- | --- |
| National Science Foundation | Faculty Early Career Development (CAREER) Award 1552419 | David Lentink |
| National Science Foundation | Graduate Research Fellowship | Ben J Hightower |
| Stanford University | Stanford Graduate Fellowship | Ben J Hightower Diana D Chin |
| Netherlands Organisation for Scientific Research | Research Program ZERO (P15-06) | Patrick WA Wijnings |
| National Defense Science and Engineering Graduate | Graduate Fellowship | Diana D Chin |

The funders had no role in study design, data collection and interpretation, or the decision to submit the work for publication.

### Author contributions

Ben J Hightower, Conceptualization, Resources, Data curation, Formal analysis, Supervision, Funding acquisition, Validation, Investigation, Visualization, Methodology, Writing - original draft, Project administration, Writing - review and editing; Patrick WA Wijnings, Conceptualization, Data curation, Formal analysis, Validation, Investigation, Visualization, Methodology, Writing - original draft, Writing - review and editing; Rick Scholte, Conceptualization, Resources, Data curation, Formal analysis, Validation, Investigation, Visualization, Methodology, Writing - original draft, Writing - review and editing; Rivers Ingersoll, Conceptualization, Resources, Investigation, Methodology, Writing - review and editing; Diana D Chin, Conceptualization, Writing - review and editing; Jade Nguyen, Conceptualization, Formal analysis, Writing - review and editing; Daniel Shorr, Conceptualization, Formal analysis, Methodology, Writing - review and editing; David Lentink, Conceptualization, Resources, Supervision, Funding acquisition, Investigation, Methodology, Project administration, Writing - review and editing

### Author ORCIDs

Ben J Hightower ⓘ https://orcid.org/0000-0003-3423-3948
Rivers Ingersoll ⓘ http://orcid.org/0000-0001-7194-0911
Diana D Chin ⓘ https://orcid.org/0000-0002-3015-7645
David Lentink ⓘ https://orcid.org/0000-0003-4717-6815

## Ethics

Animal experimentation: All bird training and experimental procedures were approved by Stanford's Administrative Panel on Laboratory Animal Care (APLAC-31426).

## Decision letter and Author response

Decision letter https://doi.org/10.7554/eLife.63107.sa1
Author response https://doi.org/10.7554/eLife.63107.sa2

# Additional files

## Supplementary files

• Supplementary file 1. Summary of the number of acoustic measurements made for each bird. To obtain frequency resolution $\leq$ 2 Hz, we selected feeding flights of 0.5 s or longer.

• Supplementary file 2. Comparison between the 10-element distributed source model and equivalent point source model. To investigate how well hummingbird flapping wing hum can be approximated with a single acoustic source per wing, we created a distributed oscillating source model with ten equally spaced elements along each wing. The force distribution was adapted from a high-fidelity model by *Ingersoll and Lentink, 2018* for the same hummingbird species. There is close agreement in magnitude for the first ten harmonics of the single and ten source model.

• Supplementary file 3. Comparison between the full and simplified acoustic models. There is reasonable agreement in magnitude for the first four harmonics.

• Supplementary file 4. Comparison between the different acoustic source locations. When the acoustic source is located at $R_3$ (chosen), $R_2$, or $R_4$, the resultant spectra have similar peak magnitudes.

• Supplementary file 5. Summary of values used for paradigm animals in acoustic models. *Culex quinquefasciatus* was adapted from *Bomphrey et al., 2017*. *Drosophila* hydei mass was adapted from *Greenewalt, 1962*, while the other parameters were adapted from *Muijres et al., 2014*. *Manduca sexta* parameters were adapted from *Zheng et al., 2013*. *Calypte anna* values were obtained from the present experiment. *Forpus coelestis* values were adapted from *Chin and Lentink, 2017*. To simplify the comparison between the five paradigm animals, we approximated the stroke plane as horizontal and the normalized lift profile to have the same shape as the reported vertically oriented force profile ('Normalized Lift Profile Proxy'), so that the lift generated during a wingbeat summed up to body weight for all associated species in the same way. ** and ***: these forces do not equate to lift, but we used the normalized profile as an approximation for the lift profile. * and ***: these forces do not necessarily equate to body weight when integrated over a wingbeat in hover. **: these forces do equate to body weight when integrated over a wingbeat in hover. * and ** and ***: the normalized profiles of these forces were used and either equate to or are a proxy for the lift profiles.

• Transparent reporting form

## Data availability

All data needed to evaluate the conclusions presented in the paper are available on Dryad, https://doi.org/10.5061/dryad.73n5tb2vs.

The following dataset was generated:

| Author(s) | Year | Dataset title | Dataset URL | Database and Identifier |
|---|---|---|---|---|
| Hightower BJ, Wijnings PWA, Scholte R, Ingersoll R, Chin DD, Nguyen J, Shorr D, Lentink D | 2020 | Data from: How Oscillating Aerodynamic Forces Explain the Timbre of the Hummingbird's Hum and Other Animals in Flapping Flight | https://doi.org/10.5061/dryad.73n5tb2vs | Dryad Digital Repository, 10.5061/dryad.73n5tb2vs |

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

## Appendix 1

### Birds

For both the 3D aerodynamic force platform (AFP) and acoustic microphone array experiments, six male Anna's hummingbirds (*Calypte anna*, *N* = 6 birds, *n* = 2 flights each) were captured using drop-door traps and released again on the same day at the location of capture. Hummingbirds were housed in individual cages before and after the experiment. Birds were deemed trained once they acclimated to their flight chamber and fed ad libitum on sugar solution from the artificial flower. All bird training and experimental procedures were approved by Stanford's Administrative Panel on Laboratory Animal Care.

### Acoustic microphone performance

The microphones used are Akustica AKU242 MEMS sensors which are sensitive to stiction: liquid, dust or other particulate matter can enter the acoustic port and cause the microphone membrane to become temporarily or permanently stuck. This can trigger the self-reset mechanism of the microphone which causes erroneous spikes in the output signal. Since the microphone captures only the dynamic pressure, the mean output value of a microphone in normal operation should always be close to zero (with a small and fixed offset due to the sensor's internal analog-digital converter). Based on this, the heuristic we use to flag microphones that may have experienced stiction during the measurement is when the average of the raw microphone data was greater than 15% of the digital full scale.

### Sound and room isolation

Sources of background noise and acoustic reflections were mitigated to achieve accurate force and acoustic recordings. Since force plates of the 3D AFP act as pressure transducers capable of measuring miniscule pressure variations, they are capable of inadvertently measuring infrasonic pressure variations due to air-conditioning systems. Both the 3D AFP and acoustic experiments were thus performed at an isolated field station. The field station has low background noise levels of 36.6 dB because it is remote and has no air conditioning system. We optimized the position of the acoustic flight arena such that it was centered from the walls (by more than 1.5 m), raised 1.5 m from the ground, and 1.5 m below the ceiling to limit the effects of acoustic reflections. Further, acoustic foam (Alphamax anechoic wedge foam, eight in thick) was placed on the ground to attenuate acoustic reflections. Both the acoustic setup and the 3D AFP were situated on three Mighty Mount M10 rubber supports (Part No. 25–2205, 80/20 Inc) for ground vibration and shock isolation.

### Calculation of lift and drag

The in vivo shoulder location, the wing chord, velocity distribution and wingtip kinematics determine the motion of $R_3$. We determined the wing velocity vector, $\boldsymbol{v}$, by taking the (component-wise) time derivative of the wing radius position vector, $\boldsymbol{r}$, in a world reference frame. Considering the velocity distribution along the radius of the hummingbird wing is linear within good approximation (*Ingersoll and Lentink, 2018*; *Ingersoll et al., 2018*), the velocity vector of the acoustic point source $\boldsymbol{v}_{R3}$ at $R_3/R$ is:

$$\boldsymbol{v}_{R3} = \frac{R_3}{R}\boldsymbol{v}_{\text{tip}}. \tag{A1}$$

The aerodynamic force generated by each wing is equal to the vector sum of the lift (with magnitude $L$ and direction $\boldsymbol{\ell}_L$) and drag (with magnitude $D$ and direction $\boldsymbol{\ell}_D$) at $R_2$, the second moment of area (*Deetjen et al., 2020*). As the AFP measures *net* forces, the forces in the longitudinal ($F_x$) and vertical ($F_z$) directions can be measured directly since the contributions from the left and right wings sum together. On the other hand, because each wingbeat is symmetric about the bird's midline, the lateral force $F_y$ from the left and right wings cancel out when measured by the AFP (*Deetjen et al., 2020*). Thus, we defined the 3D force from each wing as:

$$\boldsymbol{F}_{\text{wing}} \overset{\text{def}}{=} \begin{pmatrix} F_{x,\ \text{measured}} \\ F_{y,\ \text{calculated}} \\ F_{z,\ \text{measured}} \end{pmatrix} = -L\hat{\boldsymbol{e}}_L - D\hat{\boldsymbol{e}}_D. \tag{A2}$$

The drag unit vector is defined to act in the opposite direction as velocity at the second moment of area $R_2$:

$$\hat{\boldsymbol{e}}_D \overset{\text{def}}{=} -\frac{\boldsymbol{v}_{R2}}{\|\boldsymbol{v}_{R2}\|} = -\hat{\boldsymbol{v}}_{R2} \tag{A3}$$

The lift unit vector acts perpendicular to the wing velocity unit vector and the wing radius unit vector ($\hat{\boldsymbol{w}}$, points from the bird's right shoulder to its right wingtip) at $R_2$:

$$\hat{\boldsymbol{e}}_L = \pm \frac{-\hat{\boldsymbol{v}}_{R2} \times \hat{\boldsymbol{w}}}{\|-\hat{\boldsymbol{v}}_{R2} \times \hat{\boldsymbol{w}}\|}. \tag{A4}$$

We designated the negative sign for the left wing and the positive sign for the right wing. Based on the lift and drag unit vectors and force vector, *Equation A2* yields three coupled equations that solve for the unknown lift ($L$) and drag ($D$) magnitudes as well as the instantaneous lateral force ($\boldsymbol{F}_{y,\text{calculated}}$). The calculated lift, drag, and lateral forces are sensitive to measurement error when the vertical and horizontal components of lift and drag are near zero, which occurs at stroke reversal. To improve the calculated force accuracy at stroke reversal, we smoothed this singularity using a regularization developed by *Deetjen et al., 2020*.

## Regularization of lift and drag at stroke reversal

The calculated lateral force and aerodynamic power were sensitive to error at stroke reversal, where the vertical and horizontal components of lift and drag are near zero. We incorporated a regularization developed by *Deetjen et al., 2020*. The sensitivities arise because solving *Equation A2* requires taking the inverse of the matrix:

$$\boldsymbol{E} = \begin{pmatrix} \hat{e}_{Dx} & \hat{e}_{Lx} \\ \hat{e}_{Dz} & \hat{e}_{Lz} \end{pmatrix}. \tag{A5}$$

When $\boldsymbol{E}$ is nearly singular, the calculated forces can reach unrealistically high values due to computational limitations. Thus, we regularized the calculated force through multiplication by a weight at each instance in time:

$$W = 1 - \max\left(0, \min\left(1, \frac{\log|\det(E)| - \log c_1}{\log c_0 - \log c_1}\right)\right), \tag{A6}$$

where $c_0$ and $c_1$ are tunable constants that determine the degree of regularization. In the regularization method, when the absolute value of the denominator is below $c_0$, the weight is zero because the result is too sensitive to be used. When the absolute value of the denominator is between $c_0$ and $c_1$, the weights are between zero and one (*Figure 2—figure supplement 3*). *Chin and Lentink, 2019* reported that values of $c_0 = 0.05$ and $c_1 = 0.35$ eliminate the spikes in lateral force for parrotlets with little effect on the mid-downstroke lift and drag values. We found altering these constants had little effect on the calculated lift and drag (*Figure 2—figure supplement 4*), so we used the values reported by *Chin and Lentink, 2019*. After applying the regularization, we used Eilers' perfect smoother (*Eilers, 2003*) to smooth the lift and drag curves so the time derivatives needed to determine the acoustic pressure remain bounded when input into the acoustic model.

## Frequency mixing

The first frequency mixing stage combines the oscillating lift and drag forces from the 3D AFP measurement (*Figure 1B*), which were filtered at 180 Hz to eliminate natural frequencies in the 3D AFP setup, and the wing kinematics, which were filtered at 400 Hz. The second frequency mixing stage comes from the calculation of acoustic pressure (*Equation 1*; spectrum shown in *Figure 2—figure supplement 5E*), specifically the inner product between the aerodynamic force vector $\boldsymbol{F}_{\text{wing}}$ (spectra

shown in *Figure 2—figure supplement 5A,B*) and $r$, the vectorial distance between the flapping wing radius (spectra shown in *Figure 2—figure supplement 5C,D*) and the fixed microphone positions. The higher harmonics come from the amplitude of $\boldsymbol{F}_{\text{wing}}$ being modulated by the nonlinear position in $r$. The resulting frequency mixer embodied by the wing hum model (*Equation A2-5*; *Figure 2—figure supplement 5*) creates higher harmonics at the sum and difference of the input frequencies (*Jessop and Evans, 1976*). Finally, one of the most obvious differences between the model and measurements is the higher noise floor in the measurements due to the background noise, acoustic reflections, and microphone properties (*Gabrielson, 1995*).

## Acoustic model for hovering flight with 3D forces

The acoustics of hummingbird flights can provide valuable insight into how they generate force. The crux of the acoustic model is *Equation 1*, where the bracketed terms indicate evaluation at the emission time $t'$:

$$t' = t - \frac{|\boldsymbol{r}|}{a_o}, \tag{A7}$$

where $r$ represents the Cartesian coordinates $(x, y, z)$ measured from the inertial observer (microphone location) to the non-inertial source (the point force moving with the wing at radius $R$):

$$r = (x - R\sin\phi, y - R\cos\theta\cos\phi, z - R\sin\theta\cos\phi). \tag{A8}$$

Note that for consistency with *Lowson, 1965*, we defined the vertical direction as $x$, the front of the bird as $z$, and the right of the bird as $y$. The rotational Mach number is defined based on both wing stroke and deviation angular velocity:

$$M = \sqrt{\left(\frac{\dot{\theta}R}{a_o}\right)^2 + \left(\frac{\dot{\phi}R}{a_o}\right)^2}. \tag{A9}$$

Consequently, the 3D components of the Mach number along each Cartesian axis are defined as:

$$\boldsymbol{M} = (M\sin\theta, -M\sin\phi\cos\theta, M\cos\phi\cos\theta), \tag{A10}$$

in which $M_r$ is the component of the instantaneous convection Mach number in the direction of the observer:

$$M_r = \frac{\boldsymbol{M} \cdot \boldsymbol{r}}{|\boldsymbol{r}|}. \tag{A11}$$

The unsteady aerodynamic forces in the aeroacoustics model are based on direct in vivo measurements using the 3D AFP, as in *Equation A2*, which we used to calculate the acoustic pressure from the right wing. The resultant equation must be solved numerically since it is a function of the emission time $t'$, which is recursively defined as a function of itself. We thus established a time vector $t$ and solved for the emission time using a Newton-Raphson root finder. To obtain the acoustic pressure from the left wing with minimal computational effort, we mirrored the wing and motion across the $xz$ plane.

## Simplified acoustic model for hovering flight from vertical forces

Considering that the stroke-resolved aerodynamic forces and kinematics we measured for Anna's hummingbird are not available for other animals, we made simplifications to apply our model across species. To do this consistently, we applied the same simplifications to all animal groups, including hummingbirds. This enabled us to validate our simplifications for hummingbirds by direct comparison of the simplified and full-fledged model results. For hummingbirds, we obtained the vertical force for *Calypte anna* hummingbirds from *Ingersoll and Lentink, 2018* and approximate the lift force $\boldsymbol{L}$ from the vertical force $\boldsymbol{F}_v$ as:

$$\boldsymbol{L} \approx \boldsymbol{F}_v. \tag{A12}$$

In this approximation we assume the vertical velocity of the wing can be ignored compared to the horizontal, which is reasonable based on our validation (*Figure 2—figure supplement 6*). Since the stroke angle of a flapping wing can be represented well by harmonic motion (*Ingersoll and Lentink, 2018*), we modeled the wing element to oscillate along an arc of radius $R$ in the $yz$ plane at a constant flapping frequency. The constant wingbeat frequency, $f_w$, drives the periodic wingbeat through the following equation for the angular position of the wing:

$$\phi = \Phi_o \sin \Omega t, \tag{A13}$$

$$\Omega = 2\pi f_w, \tag{A14}$$

where $\phi = 0$ is aligned with the $y$ axis and $A_\phi$ is the wing stroke amplitude. Through substitution of *Equation A13 and A14* into the definition of Mach number, the rotational Mach number $M$ can then be written as

$$M = \frac{\phi \cdot R}{a_o} = \frac{\Omega R \Phi_o \cos \Omega t}{a_o}, \tag{A15}$$

and the associated components of the Mach number along each Cartesian axis are:

$$\boldsymbol{M} = (0, -M \sin \phi, M \cos \phi). \tag{A16}$$

The wingbeat-resolved vertical force profile and angle of attack profile were adapted from *Ingersoll and Lentink, 2018*. To calculate the associated lift and drag values we applied the quasi-steady hummingbird aerodynamic model that corroborated lift and drag coefficients from spinning wing experiments (*Kruyt et al., 2014*) as a function of angle of attack:

$$\begin{aligned} C_L &= 0.0028 + 1.1251 \cos(0.0332\alpha + 4.6963) \\ C_D &= 1.1993 + 1.0938 \cos(0.0281\alpha + 3.1277) \end{aligned} \text{ for } \alpha < 0, \tag{A17}$$

$$\begin{aligned} C_L &= 0.0031 + 1.5842 \cos(0.0301\alpha + 4.7124) \\ C_D &= 8.3171 + 8.1909 \cos(0.0073\alpha + 3.1416) \end{aligned} \text{ for } \alpha \geq 0. \tag{A18}$$

Using the wing lift, $C_L$, and drag, $C_D$, coefficient combined with the measured angle of attack, $\alpha$, the drag can be calculated based on the lift as:

$$D = L \left( \frac{C_L}{C_D} \right)^{-1}. \tag{A19}$$

Since lift acts in the vertical direction and drag acts in the x-y plane, the aerodynamic point force generated instantaneously by the wing is:

$$\boldsymbol{F} = (-L, -D \sin \phi, D \cos \phi). \tag{A20}$$

At stroke reversal, there are sharp peaks that occur in the drag curve. This is due to the extreme angle of attack transition from positive to negative (and vise versa) that occurs at stroke reversal. To mitigate the numerical discontinuity in the quasi-steady model during wingbeat reversal, the quasi-steady lift and drag curves are filtered using Eilers' perfect smoother so the time derivatives that feed into the acoustic pressure remain bounded (*Eilers, 2003*). Lastly, to calculate the lift and drag on each wing, aerodynamic symmetry was assumed, and we could thus simply divide the lift and drag predicted for the whole bird by two to calculate the force and associated acoustic radiation for each wing.

## Location of point force along wing radius

While the theoretical location of the force is at $R_3$, its location as the effective acoustic point source should be verified in practice. To determine the appropriate radial distance of the effective acoustic source, we performed a scaling analysis on *Equation 1*. This shows the dependence of the acoustic pressure distribution on wing velocity distribution, and combined with knowledge of hummingbird

morphology this validates our choice of placing the effective acoustic point source, the net aerodynamic force generated by the right wing, at $R_3$:

$$p \propto \frac{\rho_o l}{(1-M_r)^2 r} \left( \frac{U^3}{a_o} + \frac{U^4}{(1-M_r)a_o^2} + \frac{U^2 l}{r} \right). \tag{A21}$$

Thus, the acoustic pressure depends on the second, third, and fourth powers of velocity. This is equivalent to how point forces that depend on these powers of velocity are applied at the respective moment of area in blade-element models of flapping flight. Based on our analogous distributed acoustic source model for a hummingbird wing, the second, third, and fourth moments of area ($R_2$, $R_3$, and $R_4$ respectively) for calculating the associated effective acoustic point source locations are:

$$R_2/R = \sqrt{\frac{1}{S} \int_0^R c(r) r^2 dr} \approx 0.50 R, \tag{A22}$$

$$R_3/R = \sqrt[3]{\frac{1}{S} \int_0^R c(r) r^3 dr} \approx 0.55 R, \tag{A23}$$

$$R_4/R = \sqrt[4]{\frac{1}{S} \int_0^R c(r) r^4 dr} \approx 0.60 R, \tag{A24}$$

where $c(r)$ is the chord length of the wing element at radius $r$. Thus, the point of application of the force on the wing occurs at some combination of $R_2$, $R_3$, and $R_4$. At the nearfield distances of the microphones in our in vivo aeroacoustics measurements, we found a distance of 0.55 $R$ fits the data well (*Figure 2—figure supplement 7*, *Supplementary file 4*). This effective acoustic point source distance agrees with wind turbine acoustics at low frequencies (*Oerlemans et al., 2001*).

## Dimensional analysis and scaling of radiated acoustic power

We performed dimensional and scaling analysis to gain a better understanding of the importance of parameters like mass, wingspan, and flapping frequency in the production of sound. We investigated radiated acoustic power $P$, which encompasses the total sound energy radiated by a source in all directions, by integrating it over an enclosing spherical surface that includes all sources. Because of the integration, the radiated acoustic power is independent of parameters like source size. For the flapping animals we study here, the total acoustic power is acoustic intensity $I$ integrated over the surface of a sphere $S$ of a given radius that encloses them (and their unsteady aerodynamic wake) entirely:

$$P = \int_S I dS = \int_S \frac{p^2}{\rho_o a_o} dS. \tag{A25}$$

In flapping flight, the time-averaged speed of the wingtip scales as (*Lentink and Dickinson, 2009*):

$$U \cong 4\Phi_o R f_w. \tag{A26}$$

This allowed us to obtain the Mach number for flapping flight:

$$M_f = \frac{U}{a_o} \cong \frac{4\Phi_o R f_w}{a_o}. \tag{A27}$$

The Mach vector $M_i$ contains the components of the flapping Mach number along each Cartesian coordinate and thus depends on wing stroke and deviation:

$$\boldsymbol{M} = (M_f \sin\theta, -M_f \sin\phi\cos\theta, M_f \cos\phi\cos\theta) = M_f(\sin\theta, -\sin\phi\cos\theta, \cos\phi\cos\theta). \qquad (A28)$$

Since trigonometric functions are bounded by $-1$ and $1$, $M_i$ has the same order of magnitude scaling as $M_f$. If there is no deviation, $\theta = 0$, meaning $M_x = 0$. However, $M_y$ and $M_z$ will be maximized and only depend on the stroke angle since the cosine of zero is one.

Similarly, the instantaneous convective Mach number is the Mach vector in the direction of the observer:

$$M_r = \frac{\boldsymbol{M} \cdot \boldsymbol{r}}{|\boldsymbol{r}|}. \qquad (A29)$$

The vector $\boldsymbol{r}/|\boldsymbol{r}|$ has a magnitude of unity, so $M_r$ scales as $M$. The distance to the observer $\boldsymbol{r}$ is defined as:

$$\boldsymbol{r} = (x - R\sin\theta, y - R\cos\phi\cos\theta, z - R\sin\phi\cos\theta). \qquad (A30)$$

Thus, when $\boldsymbol{M}$ is dotted with $\boldsymbol{r}$ and integrated over the surface of the sphere, if one term is maximized in $M$, it will be compensated for by a commensurate change in $\boldsymbol{r}$.

Small animals tend to have higher flapping frequencies but smaller wingspans (*Shyy et al., 2013*). We plotted the flapping Mach number for all 170 animals and, as expected, found it is small compared to unity (*Figure 4—figure supplement 2*):

$$M_f \lesssim 0.1. \qquad (A31)$$

Since the flapping Mach number is less than 0.3, it is subsonic. Substituting the representative scales for a flapping wing, we derived how the time rate of change of the flapping Mach number scales:

$$\frac{\partial M_r}{\partial t} \cong \frac{U f_w}{a_o} = \frac{4\Phi_o R f_w^2}{a_o}. \qquad (A32)$$

Next, we nondimensionalized *Equation 1* by creating the following nondimensional variables (denoted by *):

$$r^* = \frac{r}{r_o}; F^* = \frac{F}{F_o}; t^* = \frac{Ut}{R} = 4\Phi_a f_w t; p^* = \frac{p}{\Delta p}; P^* = \frac{P}{P_o}, \qquad (A33)$$

where $r^*$ is the nondimensional distance from the observer (normalized by a distance $r_o$) and $F^*$ is the nondimensional force (normalized by a force scale $F_o$). Further, $t^*$ is the nondimensional time (normalized by $U$, the absolute time-averaged speed of the flapping wing at the wingtip and by $R$, the wing radius), $p^*$ is the nondimensional sound pressure (normalized by a small pressure amplitude $p$ such that $p \ll p$), and $P^*$ is the nondimensional total acoustic power (normalized by a reference power $P_o$). After which we plugged *Equation 1* into the equation for acoustic power *Equation A25* to nondimensionalize the terms in *Equation A33*:

$$P_o P^* = \int\limits_S \frac{(\Delta p p^*)^2}{\rho_o a_o} dS, \qquad (A34)$$

where:

$$\Delta p p^* = \left[ \frac{r_o r^*}{4\pi a_o |r_o r^*|^2} \left( \frac{F_o U}{R} \frac{\partial F^*}{\partial t^*} + F_o F^* \frac{U f_w}{a_o} \right) \right] + \left[ \frac{F_o F^*}{4\pi |r_o r^*|^2} \right]. \qquad (A35)$$

We algebraically simplified the above equation to separate most of the dimensional terms from the nondimensional terms:

$$\Delta p p^* = \left[ \frac{1}{4\pi a_o r_o} \frac{1}{r^*} \left( \frac{F_o U}{R} \frac{\partial F^*}{\partial t^*} + \frac{F_o U f_w}{a_o} F^* \right) \right] + \left[ \frac{F_o}{4\pi r_o^2} \frac{F^*}{r^{*2}} \right], \qquad (A36)$$

$$\Delta pp^* \left( \frac{F_o}{4\pi r_o} \frac{U}{a_o} \frac{1}{R} \frac{1}{r^*} \right)^{-1} = \frac{\partial F^*}{\partial t^*} + \frac{Rf_w}{a_o} F^* + \left( \frac{U}{a_o} \frac{1}{R} \right)^{-1} \frac{1}{r_o} \frac{F^*}{r^*}, \tag{A39}$$

$$\Delta pp^* = \frac{F_o}{4\pi r_o} \frac{1}{r^*} \left[ \frac{U}{a_o} \left( \frac{1}{R} \frac{\partial F^*}{\partial t^*} + \frac{f_w}{a_o} F^* \right) + \frac{1}{r_o} \frac{F^*}{r^*} \right], \tag{A37}$$

$$\Delta pp^* = \frac{F_o}{4\pi r_o} \frac{U}{a_o} \frac{1}{R} \frac{1}{r^*} \frac{\partial F^*}{\partial t^*} + \frac{F_o}{4\pi r_o} \frac{U}{a_o} \frac{f_w}{a_o} \frac{F^*}{r^*} + \frac{F_o}{4\pi r_o^2} \frac{F^*}{r^{*2}}, \tag{A38}$$

$$\Delta pp^* = \left( \frac{F_o}{4\pi r_o} \frac{U}{a_o} \frac{1}{R} \frac{1}{r^*} \right) \left[ \frac{\partial F^*}{\partial t^*} + \frac{Rf_w}{a_o} F^* + \frac{a_o R}{U r_o} \frac{F^*}{r^*} \right]. \tag{A40}$$

We plugged this nondimensional representation of sound pressure into *Equation A25* to solve for the nondimensional radiated acoustic power:

$$P_o P^* = \left( \frac{F_o U}{4\pi r_o R r^*} \frac{1}{\sqrt{\rho_o a_o^3}} \right)^2 \int_S \left[ \frac{\partial F^*}{\partial t^*} + \frac{Rf_w}{a_o} F^* + \frac{a_o R}{U r_o} \frac{F^*}{r^*} \right]^2 dS. \tag{A41}$$

We made the substitution $U \cong 4\Phi_o R f_w$ (*Lentink and Dickinson, 2009*) and simplified algebraically:

$$P_o P^* = \left( \frac{F_o \Phi_o f_w}{\pi r_o r^*} \frac{1}{\sqrt{\rho_o a_o^3}} \right)^2 \int_S \left[ \frac{\partial F^*}{\partial t^*} + \frac{Rf_w}{a_o} F^* + \frac{a_o}{4\Phi_o f_w r_o} \frac{F^*}{r^*} \right]^2 dS. \tag{A42}$$

We set all dimensionless variables equal to their order of magnitude one:

$$P_o = \left( \frac{F_o \Phi_o f_w}{\pi r_o} \frac{1}{\sqrt{\rho_o a_o^3}} \right)^2 \int_S \left[ 1 + \frac{Rf_w}{a_o} + \frac{a_o}{4\Phi_o f_w r_o} \right]^2 dS. \tag{A43}$$

The first term in *Equation A43* dominates the second term in the scaling analysis because they differ by the following multiplicative factor much smaller than one:

$$\frac{Rf_w}{a_o}. \tag{A44}$$

This factor is small because flapping wing animals act as compact acoustic sources. Plotting this factor for the 170 animals we selected shows it is small compared to unity: $Rf_w/a_o \lesssim 0.01$ (*Figure 4—figure supplement 1*). Note that at higher harmonics, this factor is no longer small compared to unity, limiting our analysis to the first 10 harmonics. Since the Mach numbers for flapping flight were small compared to unity as demonstrated earlier, this multiplicative factor was also small compared to unity when considering the fundamental flapping frequency and can thus be neglected:

$$P_o = \int_S \left[ \frac{F_o \Phi_o f_w}{\pi r_o} \frac{1}{\sqrt{\rho_o a_o^3}} + \frac{F_o}{\pi r_o^2} \frac{1}{\sqrt{\rho_o a_o^3}} \frac{a_o}{4} \right]^2 dS, \tag{A45}$$

$$P_o = \left( \frac{F_o}{\pi \sqrt{\rho_o a_o^3}} \right)^2 \int_S \left[ \frac{\Phi_o f_w}{r_o} + \frac{1}{r_o^2} \frac{a_o}{4} \right]^2 dS. \tag{A46}$$

In the limit of large $r_o$ the second term in the integrand is negligible, yielding:

$$P_o = \left( \frac{F_o}{\pi \sqrt{\rho_o a_o^3}} \right)^2 \int_S \left[ \frac{\Phi_o f_w}{r_o} \right]^2 dS. \tag{A47}$$

For a sphere of radius $r_o$, the integrand can be evaluated as:

$$P_o = \left(\frac{F_o}{\pi\sqrt{\rho_o a_o^3}}\right)^2 \left[\frac{\Phi_o f_w}{r_o}\right]^2 (4\pi r_o^2), \tag{A48}$$

and simplified as:

$$P_o = \left(\frac{2F_o\Phi_o f_w}{\sqrt{\pi\rho_o a_o^3}}\right)^2 = \frac{4F_o^2\Phi_o^2 f_w^2}{\pi\rho_o a_o^3}. \tag{A49}$$

For an animal hovering in equilibrium, the sum of the vertical aerodynamic force (as in *Figure 4A*) generated by both wings should equal the animal's weight, $mg$, stroke-averaged. To investigate how the radiated acoustic power scales with mass, we can thus substitute $F_o = mg$:

$$P_o \propto \frac{4m^2 g^2 \Phi_o^2 f_w^2}{\pi\rho_o a_o^3}. \tag{A50}$$

Consequently, $P_o \propto m^2$, which shows logarithmic plots of radiated acoustic power as a function of mass have an ideal slope of 2.0 if all assumptions are met (*Figure 4E*).

We also plotted $P_o$ versus $\frac{4F_o^2\Phi_o^2 f_w^2}{\pi\rho_o a_o^3}$, which yielded an ideal slope of 1.0, corroborating our simulation results over the 170 different animals (*Figure 4F*).

