## [Decision Letter]

**Acceptance summary:**

In this manuscript, Hightower et al. detail the spatiotemporal acoustic emissions of a hummingbird in flight. Hummingbirds are an excellent system for an experimental study of wing-emitted acoustics because they can be trained to hover in place, which allows unhindered measurement of their aerodynamic forces, wing movement, and sound emission. The paper's main strength is in putting together three approaches: force measurement using a previously described force platform method; wing kinematics using high-speed videography; and acoustic measurements using 2176 MEMS microphones. Using these measurements, the authors put together a general model of acoustic emissions from birds and insects, shedding light on this complex bioacoustic behaviour.

**Decision letter after peer review:**

Thank you for submitting your article "How Oscillating Aerodynamic Forces Explain the Timbre of the Hummingbird's Hum and Other Animals in Flapping Flight" for consideration by *eLife*. Your article has been reviewed by three peer reviewers, one of whom is a member of our Board of Reviewing Editors, and the evaluation has been overseen by Christian Rutz as the Senior Editor. The reviewers have opted to remain anonymous.

The reviewers have discussed their reviews with one another, and the Reviewing Editor has drafted this decision letter to help you prepare a revised submission. Overall, the reviewers were very positive about the work, but they have a few comments that they would like you to address before recommending acceptance of the manuscript at *eLife*.

Summary:

In this manuscript, Hightower et al. detail the spatio-temporal acoustic emissions of a hummingbird in flight. Hummingbirds are an excellent system for an experimental study of wing-emitted acoustics, because they can be trained to hover in place, which allows unhindered measurement of their aerodynamic forces, wing movement and sound emission. The paper's main strength is in putting together three approaches: force measurement using a previously described force platform method; wing kinematics using high speed videography; and acoustic measurements using 2176 MEMS microphones. Using these measurements, the authors put together a general model of acoustic emissions from birds and insects, shedding light on this complex bioacoustic behavior.

Essential revisions:

1) The authors start with the general statement that "the source of the hummingbird's distinctive hum is not understood" and proceed to show that the source is the acoustic emissions resulting from their beating wings. The reviewers were not sure what the term "not understood" is supposed to mean. This is not the first paper on acoustic emissions of flapping wings by any means – in many ways, this paper recalls a previous numerical study by Seo, Hedrick and Mittal, 2019, which investigated the tone in mosquitoes. The theoretical structure and modeling used there is similar to the one in this paper – and both derive from the classic studies of Ffowcs Williams and Hawkings (this original paper, although mentioned, is oddly enough not cited here). The reviewers felt that this was not sufficiently acknowledged in this manuscript. Instead of cursory statements like "in vivo evidence is lacking" or "no simple first principles model exists…", it is important to emphasize what specifically is known and what is not known about how the various harmonics contribute and combine to produce a distinct sound, and which harmonic contains more energy etc. The final model calculates the relative distribution of energies in the various frequencies, thus allowing a more precise estimation of which harmonic would contribute the most to the emitted sound.

2) The experiments are sufficiently rigorous and build nicely upon previous methods developed in the Lentink lab. Based on these measurements, the authors have developed what they call "a first-principles' model" – although the reviewers did not agree with this description, because the model described here is built on top of a classic model by Ffowcs Williams and Hawkings. The term "first principles" suggests that it was built from basic concepts, which is misleading and quite unnecessary. Once validated, such a model is useful because it can be extended to other flapping insects and birds.

3) In general, the Results were very difficult to read because the authors use a lot of technical jargon in their main text which could be in the Materials and methods instead. The Results could be written in a way so as to avoid sentences such as "Each frequency component of the holograms was regularized independently using a Bayesian evidence method (Weis-Fogh, 1973) before adding them all together to create the broadband NAH results. To reduce distortions due to frequency leakage, linear predictive border padding (Kruyt et al., 2014; Oerlemans et al., 2001) was applied to the time signals." In writing such sentences the authors assume familiarity with the Bayesian Evidence Method or the Linear Predictive border padding or even Blade-Element theory – it is unlikely that the broad *eLife* readership would be familiar with all of these. De-jargonifying these types of sections is a very important revision that the reviewers expect.

4) It was not clear how well the assumption of a point source of sound holds if the microphones are placed at close distance to the hovering bird. For a small animal like a mosquito, this assumption seems reasonable, but not if the source of sound is more extended. This point may prove especially crucial when the model is scaled up for larger birds.

5) Related to the above, how well does the assumption of a point source of sound hold in the comparison between the measurement and the model for hummingbirds – and to what extent is the mismatch between the records attributable to this assumption?

6) At least some of the higher harmonics may occur due to events that occur multiple times during a single wing stroke. For instance, the effect of the flexible wing moving back and forth causes the wingtip to flutter at frequencies that are higher harmonic of the wing motion (e.g., Sane and Jacobson, 2006). The authors do not discuss this possibility.

7) Lift and drag are merely two components of the same force. Hence, it is not immediately clear to the reviewers why most "frequency content" for lift is in 1st and 2nd harmonic whereas for drag, it is in 2nd and 3rd harmonic.

8) On the issue of directional radiation of sound, the reviewers would have liked to see a comparison between this study and previous numerical ones (e.g., Seo et al.) that have explored this.

9) The reviewers were not sure if the authors' explanation of the hummingbird "cobra" manoeuvre is quite correct. According to the Hunter and Picman, 2005 paper, the cobra manoeuvre is accompanied by a trill pulse, to generate which they must increase their flapping frequency. Mechanistically, this is entirely different than the mechanism that the authors have described in which the increase in flapping frequency enhances the acoustic pressure by a moderate amount, rather than create a different sound. Can the authors please double check whether their explanation is indeed valid? In general, it felt like this last section was somewhat speculative, but the evidence did not quite fit.

---

## [Author Response]

Essential revisions:1) The authors start with the general statement that "the source of the hummingbird's distinctive hum is not understood" and proceed to show that the source is the acoustic emissions resulting from their beating wings. The reviewers were not sure what the term "not understood" is supposed to mean. This is not the first paper on acoustic emissions of flapping wings by any means – in many ways, this paper recalls a previous numerical study by Seo, Hedrick and Mittal, 2019, which investigated the tone in mosquitoes. The theoretical structure and modeling used there is similar to the one in this paper – and both derive from the classic studies of Ffowcs Williams and Hawkings (this original paper, although mentioned, is oddly enough not cited here). The reviewers felt that this was not sufficiently acknowledged in this manuscript. Instead of cursory statements like "in vivo evidence is lacking" or "no simple first principles model exists…", it is important to emphasize what specifically is known and what is not known about how the various harmonics contribute and combine to produce a distinct sound, and which harmonic contains more energy etc. The final model calculates the relative distribution of energies in the various frequencies, thus allowing a more precise estimation of which harmonic would contribute the most to the emitted sound.

Clarified; we replaced our cursory statements with more detail. We contextualize our contributions with regards to previous work by Seo et al. and Ffowcs Williams and Hawkings. Our contributions include the novel experimental setup on an animal that has not been studied acoustically in this manner, whereby we use microphones to directly measure a ground-truth for the produced sound. Our model is also different, as we use measured force inputs into a point model with low computational costs compared to using measured kinematics as inputs to a complex model that uses CFD for the entire wing (Seo et al.).

2) The experiments are sufficiently rigorous and build nicely upon previous methods developed in the Lentink lab. Based on these measurements, the authors have developed what they call "a first-principles' model" – although the reviewers did not agree with this description, because the model described here is built on top of a classic model by Ffowcs Williams and Hawkings. The term "first principles" suggests that it was built from basic concepts, which is misleading and quite unnecessary. Once validated, such a model is useful because it can be extended to other flapping insects and birds.

We changed “first-principles model” to “idealized model,” since our model is idealized in the sense that it assumes the wings are rigid airfoils, thereby neglecting auxiliary effects such as wingtip flutter, feather whistle, and (turbulent) vortex dynamics. This better communicates the rigor of our approach.

3) In general, the Results were very difficult to read because the authors use a lot of technical jargon in their main text which could be in the Materials and methods instead. The Results could be written in a way so as to avoid sentences such as "Each frequency component of the holograms was regularized independently using a Bayesian evidence method (Weis-Fogh, 1973) before adding them all together to create the broadband NAH results. To reduce distortions due to frequency leakage, linear predictive border padding (Kruyt et al., 2014; Oerlemans et al., 2001) was applied to the time signals." In writing such sentences the authors assume familiarity with the Bayesian Evidence Method or the Linear Predictive border padding or even Blade-Element theory – it is unlikely that the broad eLife readership would be familiar with all of these. De-jargonifying these types of sections is a very important revision that the reviewers expect.

We moved the specifics of the NAH processing and the instrumentation to the Materials and methods section and removed the jargon regarding Blade-Element theory. Thank you for helping us make our research more widely accessible.

4) It was not clear how well the assumption of a point source of sound holds if the microphones are placed at close distance to the hovering bird. For a small animal like a mosquito, this assumption seems reasonable, but not if the source of sound is more extended. This point may prove especially crucial when the model is scaled up for larger birds.

Clarified; we added more detail about the compact acoustic simplification, which allows one to neglect the retarded time differences between different parts of the wing surface that emit sound. This is also a crucial part of our model as we assume the wing is acoustically transparent and does not scatter sound over itself. Figure 4—figure supplement 2 shows all animals used in our extended analysis have Rfw/ao≲0.01, and thus act as compact acoustic sources whereby the wavelength of the sound generated is two orders of magnitude larger than the wing radius.

Furthermore, since wing length is inversely proportional to flapping frequency (Greenewalt, 1962), the assumption of acoustic compactness also holds for most other species than those used in our analysis. This means that concentrating the radial force distribution of the wing into an equivalent point force source is a valid assumption across a wide range of flying animals: for equal compactness (and equivalent kinematics), model error due to the point source assumption at one wavelength away from an animal is comparable to the hummingbird case (see below). For example, hummingbirds are as acoustically compact as mosquitos, with a value of 0.007 for hummingbirds and 0.006 for mosquitos.

5) Related to the above, how well does the assumption of a point source of sound hold in the comparison between the measurement and the model for hummingbirds – and to what extent is the mismatch between the records attributable to this assumption?

Clarified; we created a distributed source model with 10 equally spaced elements along the wing (with force distribution according to Ingersoll and Lentink, 2018) to compare to the single source model (Figure 2—figure supplement 1A-B, Supplementary file 2). The median difference between the point force and distributed model is 0.1 dB (Figure 2—figure supplement 1A), and the holograms throughout the wingbeat are indistinguishable (Figure 2—figure supplement 1B), indicating the point source model is a reasonable assumption. Other differences between the model and measurement can be attributed to effects like wingtip flutter, acoustic scattering by the wing and body of the hummingbird, and turbulent effects. The combined effect of these differences is bounded by the differences in the spectra (Figure 2A) and is limited to ~7 dB or less (maximum difference between the model and the measurement ± 1 SD).

6) At least some of the higher harmonics may occur due to events that occur multiple times during a single wing stroke. For instance, the effect of the flexible wing moving back and forth causes the wingtip to flutter at frequencies that are higher harmonic of the wing motion (e.g., Sane and Jacobson, 2006). The authors do not discuss this possibility.

The effects of wing flutter are present in the actual microphone measurements but not in the simulations. Therefore, their effect is bounded by the differences in Figure 2A. Due to the good agreement between simulation and experiment, the contributions of the flutter effects and other secondary effects not captured by the simulation are limited to ~7 dB or less. Conversely, the presence of these effects could explain some of the remaining differences between the measurements and simulation, especially at higher harmonics. Clarification has been added to the Results section.

7) Lift and drag are merely two components of the same force. Hence, it is not immediately clear to the reviewers why most "frequency content" for lift is in 1st and 2nd harmonic whereas for drag, it is in 2nd and 3rd harmonic.

Clarified; lift and drag are two components of the same force that have different directions. Drag is in the direction opposite wing motion, and lift is orthogonal to the wing radius vector and drag direction (Figure 1B). In the *drag direction*, the force primarily fluctuates twice per wingbeat, associated with the drag generated during upstroke and downstroke (Figure 1C). In the *lift direction*, the force primarily fluctuates once per wingbeat, corresponding to the higher forces generated during downstroke versus upstroke (Figure 1C). Clarification and simplification have been added to the Figure 1 legend and Results section.

8) On the issue of directional radiation of sound, the reviewers would have liked to see a comparison between this study and previous numerical ones (e.g., Seo et al.) that have explored this.

We include a comparison between the sagittal plane, for which Seo et al., 2019, provided for a mosquito, with the directivity results we obtained for a hummingbird. In comparison with the results from Seo et al., the first and fourth harmonic are approximately orthogonal, with the acoustic directivity of the hummingbird pointed aft in the first harmonic and forward in the fourth harmonic. The second harmonic differs by ~20°, with the acoustic directivity of the hummingbird directed more upright, while the third harmonic differs by ~20° with the hummingbird directed aft. We note that this is expected considering the wingbeat kinematics of hummingbirds and mosquitos differ substantially. Mosquitos have a shallower stroke amplitude, a more horizontally oriented stroke plane, and a higher wingbeat frequency.

9) The reviewers were not sure if the authors' explanation of the hummingbird "cobra" manoeuvre is quite correct. According to the Hunter and Picman, 2005 paper, the cobra manoeuvre is accompanied by a trill pulse, to generate which they must increase their flapping frequency. Mechanistically, this is entirely different than the mechanism that the authors have described in which the increase in flapping frequency enhances the acoustic pressure by a moderate amount, rather than create a different sound. Can the authors please double check whether their explanation is indeed valid? In general, it felt like this last section was somewhat speculative, but the evidence did not quite fit.

We removed sections regarding the hummingbird “cobra” maneuver to limit speculation.